# Breaking through Data Scarcity: Knowledge Transfer in Offline Reinforcement Learning

## Abstract

We focus on knowledge transfer in offline reinforcement learning (RL), which aims to significantly improve the learning of an optimal policy in a target task based on a pre-collected dataset without further interactions with the environment. Data scarcity and high-dimensional feature spaces seriously pose challenges to offline RL in many real-world applications, and knowledge transfer offers a promising solution. We propose a novel and comprehensive knowledge transfer framework for offline RL, which carefully considers the relationship between the target and source tasks within the linear Markov decision process (MDP) framework. This enables efficient knowledge transfer from related source tasks to enhance learning in the target task and effectively address data scarcity concerns in offline RL. Our main contributions include establishing a relationship with the learning process between the target task and source task, introducing an effective and robust knowledge transfer technique to reduce the suboptimality of the learned policy, and demonstrating the significant effectiveness of the knowledge transfer framework through detailed theoretical analysis. Our work significantly contributes to the advancement of offline RL by providing a practical and robust framework for knowledge transfer facilitating more efficient and effective data utilization in various applications.

## 1 Introduction

The reinforcement learning (RL) has achieved remarkable success in various applications, which largely relies on two crucial factors: (i) powerful function approximators, such as deep neural networks (LeCun et al., 2015; Mnih et al., 2015; Silver et al., 2016; Vinyals et al., 2017), that can approximate policies and values with high precision, and (ii) efficient data generators, like simulation environments (Bellemare et al., 2013; Todorov et al., 2012), that facilitate the collection of large amounts of data through interactions with the environment. However, in many real-world scenarios, such as robotics and healthcare, obtaining massive interactive data can be extremely costly, time-consuming, and even pose risks. Therefore, we focus on offline RL, which aims to learn an optimal policy based on a pre-collected dataset without further interactions with the environment.

In today's rapidly evolving technological landscape, offline reinforcement learning (RL) has emerged as a crucial area of research in data-driven decision-making. It aims to learn optimal policies based on datasets collected a priori, without the need for further interactions with the environment. This is particularly relevant in various domains, such as marketing, healthcare, and education, where data scarcity and high-dimensional feature spaces pose significant challenges. Unlike online RL, offline RL is still relatively less understood from a theoretical perspective (Lange et al., 2012; Levine et al., 2020), which poses significant challenges in developing reliable algorithms for practical applications. In particular, since active interactions with the environment are not feasible in offline RL, it becomes difficult to exploit the dataset without further exploration fully. Due to the lack of continuous exploration, any algorithm for offline RL may suffer from the problem of insufficient dataset coverage (Wang et al., 2020). Specifically, two main challenges arise (i) the intrinsic uncertainty, where the dataset may fail to cover the trajectory induced by the optimal policy, which contains essential information; (ii) the spurious correlation, meaning that the dataset may accidentally cover a trajectory that is unrelated to the optimal policy, but which can mislead the learned policy (Fujimoto et al., 2019; Agarwal et al., 2020; Fu et al., 2020; Gulcehre et al., 2020).

In the context of offline RL, knowledge transfer offers a promising approach to improving learning efficiency and performance. By transferring knowledge from related source tasks to a target task, we can exploit the relationship between target and source tasks to overcome the data scarcity issue. However, existing literature on knowledge transfer or transfer learning for RL lacks a thorough examination of the theoretical guarantees for value function estimation.

In this paper, we aim to answer the following question:

*Can we develop a knowledge transfer framework that effectively addresses the data scarcity and provides rigorous theoretical guarantees?*

In conclusion, we propose a novel framework for knowledge transfer in offline RL (KT-RL). The contributions of our work are concluded as follows:

1. Knowledge Transfer Framework Innovation

   - **Breaking Conventional Assumptions**: We assume that the target data is a linear combination of source data. This assumption provides a novel perspective and method for knowledge transfer in offline RL, departing from the common practices in existing literature.
   - **Comprehensive Consideration of Task Relationships**: Based on the linear Markov Decision Process (MDP) framework, we comprehensively consider the relationships between the target task and source tasks. This approach effectively addresses the data scarcity problem and enhances the learning performance of the target task.

2. Theoretical Contributions

   - **Establishing Theoretical Relationships**: We establish a theoretical relationship between the learning processes of the target task and source tasks. By introducing knowledge transfer techniques, we can reduce the suboptimality of the learned policy.
   - **Providing Bounds for Algorithm Evaluation**: Through theoretical analysis, we provide an upper bound on the suboptimality of our algorithm. Additionally, we prove the minimax optimality of the algorithm, which offers a solid basis for evaluating the performance of the algorithm.

3. Algorithm Design Contributions

   - Efficient Source Data Processing and Integration

     - **Separate Calculation of Source Data Statistics**: For each source task, we calculate statistical quantities separately. In each step, we define the empirical mean squared Bellman error (MSBE) and calculate the estimated Bellman operator, confidence bound, value function, action-value function, etc. This process fully considers the characteristics of each source task and retains its unique information. Unlike the transfer learning methods in (Chen et al., 2024; Lei et al., 2024), which require aggregating raw data from various sources, our approach only necessitates the sharing of statistical quantities from the model. From this perspective, our algorithm enhances privacy preservation by decentralized sensitive raw data.
     - **Integration of Source Data for Target Task**: We integrate the statistical quantities obtained from each source task to calculate the target task. This integration method takes into account the diversity of source data and effectively transfers knowledge. It enhances the flexibility and effectiveness of the algorithm in handling different source-target task relationships, thus improving the accuracy and effectiveness of knowledge transfer.
   - **Ensuring Dataset Compliance and Trajectory Independence**: In data collection process, we assume that the dataset complies with the underlying MDP and the trajectories are independent. This assumption simplifies the analysis process and ensures the reliability and stability of the algorithm in practical applications.

These contributions together significantly advance the field of offline RL and offer potential for more efficient and effective decision-making in various real-world applications.

## 2 RELATED WORK

In the field of offline reinforcement learning, numerous studies have been conducted to address various challenges. Our work is closely related to the following lines of research:

**Reinforcement Learning**: There is a rich body of literature on offline RL algorithms, such as (Fujimoto et al., 2019; Agarwal et al., 2020; Fu et al., 2020; Gulcehre et al., 2020). These algorithms aim to learn an optimal policy based on a pre-collected dataset without further interactions with the environment. Our proposed algorithm contributes to this area by incorporating transfer learning to enhance learning efficiency. There are several work falls within the realm of batch reinforcement learning, where the goal is to learn an optimal policy from a fixed dataset without further interactions with the environment (Shi et al., 2022; Yan et al., 2022; Li et al., 2024).

**Knowledge Transfer in Reinforcement Learning**: Numerous studies have explored transfer learning in the context of reinforcement learning. Chen et al. (2022) investigated the transfer of Q-learning, while Agarwal et al. (2023) focused on the benefits of representational transfer in reinforcement learning. These works provide valuable insights into how knowledge can be transferred between tasks in the RL domain. Transfer learning in RL aims to leverage data from related source tasks to enhance the learning on a target task (Agarwal et al., 2023). Additionally, our work is related to the broader topic of knowledge transfer in sequential decision-making. Previous studies have investigated utilizing data from existing ventures to navigate high-dimensional feature spaces and address data scarcity in new ventures (Liu, 2023; Komorowski et al., 2018; Rafferty et al., 2016). We extend this idea to the context of offline reinforcement learning, demonstrating how knowledge transfer can be applied to improve the learning efficiency in this domain. In contrast to the existing literature on transfer learning (Bastani, 2021; Lei et al., 2024; Li et al., 2022; 2023; Bastani et al., 2022; Tian & Feng, 2023), which typically assumes that source data closely resembles target data, our approach diverges from this assumption.

**Linear MDPs and High-Dimensional Feature Spaces in RL**: Dealing with high-dimensional feature spaces is a crucial challenge in offline reinforcement learning. Some works (Bellemare et al., 2013; Todorov et al., 2012) have focused on developing efficient data generators and function approximators to address this issue. Our approach builds upon these ideas by proposing a novel transfer learning framework that specifically takes into account the high-dimensional feature spaces in the source and target tasks. Additionally, linear MDPs have been studied in various RL papers (Yang & Wang, 2019; Jin et al., 2020). These studies have shown that linear MDPs can provide a useful framework for analyzing and solving RL problems. In our work, we also utilize the concept of linear MDPs to define the task discrepancy and establish the relationship between target and source tasks and the learning process in the target task.

## 3 PRELIMINARIES

In this section, we first introduce the episodic Markov decision process (MDP).

We consider an episodic MDP $(\mathcal{S}, \mathcal{A}, H, \mathcal{P}, r)$ with the state space $\mathcal{S}$, action space $\mathcal{A}$, horizon $H$, transition kernel $\mathcal{P} = \{\mathcal{P}_h\}_{h=1}^H$, and reward function $r = \{r_h\}_{h=1}^H$. We assume the reward function is bounded, that is, $r_h \in [0,1]$ for all $h \in [H]$. For any policy $\pi = \{\pi_h\}_{h=1}^H$, we define the value function $V_h^\pi : \mathcal{S} \to \mathbb{R}$ at each step $h \in [H]$ and the action-value function (Q-function) $Q_h^\pi : \mathcal{S} \times \mathcal{A} \to \mathbb{R}$ at each step $h \in [H]$ as:

$$V_h^\pi(x) = \mathbb{E}_\pi \left[ \sum_{i=h}^H r_i(s_i, a_i) \mid s_h = x \right], Q_h^\pi(x, a) = \mathbb{E}_\pi \left[ \sum_{i=h}^H r_i(s_i, a_i) \mid s_h = x, a_h = a \right]. \quad (1)$$

Here the expectation $\mathbb{E}_\pi$ in Equations (1) is taken for the randomness of the trajectory induced by $\pi$, which is obtained by taking action $a_i \sim \pi_i(\cdot \mid s_i)$ at the state $s_i$ and observing the next state $s_{i+1} \sim \mathcal{P}_i(\cdot \mid s_i, a_i)$ at each step $i \in [H]$. Meanwhile, we fix $s_h = x \in \mathcal{S}$ and $(s_h, a_h) = (x, a) \in \mathcal{S} \times \mathcal{A}$ in Equation (1). By the definition in Equations (1), we have the Bellman equation:

$$V_h^\pi(x) = \langle Q_h^\pi(x, \cdot), \pi_h(\cdot \mid x) \rangle_{\mathcal{A}}, \quad Q_h^\pi(x, a) = \mathbb{E}\left[ r_h(s_h, a_h) + V_{h+1}^\pi(s_{h+1}) \mid s_h = x, a_h = a \right],$$

where $\langle \cdot, \cdot \rangle_{\mathcal{A}}$ is the inner product over $\mathcal{A}$, while $\mathbb{E}$ is taken for the randomness of the immediate reward $r_h(s_h, a_h)$ and next state $s_{h+1}$. For any function $f : \mathcal{S} \to \mathbb{R}$, we define the transition operator at each step $h \in [H]$ and the Bellman operator at each step $h \in [H]$ as:

$$(\mathbb{P}_h f)(x, a) = \mathbb{E}\left[ f(s_{h+1}) \mid s_h = x, a_h = a \right], \quad (2)$$

$$(\mathbb{B}_h f)(x,a) = \mathbb{E}[r_h(s_h,a_h) + f(s_{h+1}) \mid s_h = x, a_h = a] = \mathbb{E}[r_h(s_h,a_h) \mid s_h = x, a_h = a] + (\mathbb{P}_h f)(x,a). \tag{3}$$

For the episodic MDP $(\mathcal{S}, \mathcal{A}, H, \mathcal{P}, r)$, we use $\pi^*$, $Q_h^*$, and $V_h^*$ to denote the optimal policy, optimal Q-function, and optimal value function, respectively. We have $V_{H+1}^* = 0$ and the Bellman optimality equation as:

$$V_h^*(x) = \max_{a \in \mathcal{A}} Q_h^*(x,a), \quad Q_h^*(x,a) = \left(\mathbb{B}_h V_{h+1}^*\right)(x,a).$$

Meanwhile, the optimal policy $\pi^*$ is specified by

$$\pi_h^*(\cdot \mid x) = \arg\max_{\pi_h} \langle Q_h^*(x,\cdot), \pi_h(\cdot \mid x) \rangle_{\mathcal{A}}, \quad V_h^*(x) = \langle Q_h^*(x,\cdot), \pi_h^*(\cdot \mid x) \rangle_{\mathcal{A}},$$

where the maximum is taken over all functions mapping from $\mathcal{S}$ to distributions over $\mathcal{A}$. We aim to learn a policy that maximizes the expected cumulative reward. Correspondingly, we define the performance metric as:

$$\text{SubOpt}(\pi; x) = V_1^{\pi^*}(x) - V_1^\pi(x), \tag{4}$$

which is the suboptimality of the policy $\pi$ given the initial state $s_1 = x$.

## 3.1 LINEAR MDP

We study the knowledge transfer for offline RL in a concrete setting: the linear MDP. We define the linear MDP following the works Yang & Wang (2019); Jin et al. (2020), where the transition kernel and expected reward function are linear in a feature map.

**Definition 1** (Linear MDP). *We say an episodic* $\text{MDP}(\mathcal{S}, \mathcal{A}, H, \mathcal{P}, r)$ *is a linear MDP with a known feature map* $\phi : \mathcal{S} \times \mathcal{A} \rightarrow \mathbb{R}^d$ *if there exist d unknown measures* $\boldsymbol{\mu}_h = \left(\mu_h^{(1)}, \ldots, \mu_h^{(d)}\right)$ *over* $\mathcal{S}$ *and an unknown vector* $\boldsymbol{\theta}_h \in \mathbb{R}^d$ *such that*

$$\mathcal{P}_h\left(x' \mid x, a\right) = \langle \phi(x,a), \boldsymbol{\mu}_h(x') \rangle, \quad \mathbb{E}\left[r_h(s_h,a_h) \mid s_h = x, a_h = a\right] = \langle \phi(x,a), \boldsymbol{\theta}_h \rangle. \tag{5}$$

*for all* $(x, a, x') \in \mathcal{S} \times \mathcal{A} \times \mathcal{S}$ *at each step* $h \in [H]$. *Here we assume* $\|\phi(x,a)\| \leq 1$ *for all* $(x, a) \in \mathcal{S} \times \mathcal{A}$ *and* $\max\{\|\boldsymbol{\mu}_h(\mathcal{S})\|, \|\boldsymbol{\theta}_h\|\} \leq \sqrt{d}$ *at each step* $h \in [H]$, *where with an abuse of notation, we define* $\|\boldsymbol{\mu}_h(\mathcal{S})\| = \int_{\mathcal{S}} \|\boldsymbol{\mu}_h(x)\| \,\mathrm{d}x$.

In the subsequent section, we propose our algorithm (Algorithm 1), which utilizes knowledge transfer to construct $\widehat{\mathbb{B}}_h \widehat{V}_{h+1}$, $\Gamma_h$, and $\widehat{V}_h$ based on the dataset $\mathcal{D} = \{(x_h^\tau, a_h^\tau, r_h^\tau)\}_{\tau,h=1}^{K,H}$. Specifically, for $\widehat{\mathbb{B}}_h \widehat{V}_{h+1}$, we build it based on $\mathcal{D}$ as follows. Recall that $\widehat{\mathbb{B}}_h \widehat{V}_{h+1}$ is intended to approximate $\mathbb{B}_h \widehat{V}_{h+1}$, where $\mathbb{B}_h$ is the Bellman operator defined in Equation (3).

## 4 PROBLEM FORMULATION

In the previous sections, we have introduced our research's background and related concepts. This section focuses on the problem formulation of knowledge transfer for offline reinforcement learning.

**The Target and Source RL Data.** Transferred RL aims to improve the learning on a target RL task by leveraging data from related source RL tasks. We consider the case where we have abundant source data from offline observational data or simulated data, while the target task only has limited offline data. Specifically, we have a target task and $L$ source tasks, which are characterized by MDPs $\mathcal{M}^{(l)} = \{\mathcal{S}, \mathcal{A}, H, \mathcal{P}^{(l)}, r^{(l)}\}$ for $l \in \{0\} \cup [L]$. The target RL task of interest is referred to as the 0-th task and denoted by a superscript " (0)," while the source RL tasks are denoted by a superscript " (l)," for $l \in [L]$.

Many existing knowledge transfer methods rely on leveraging information from source data that closely resembles the target data (Chen et al., 2022; Bastani et al., 2022; Li et al., 2022; Lei et al., 2024; Tian & Feng, 2023). However, this approach often overlooks valuable knowledge that may be present in different yet potentially related samples. Different from existing literature which imposes similarity constraints on target data and source data, we make the following assumption on source data in Assumption 1: the target data is a linear combination of source data.

**Assumption 1.** *For $l \in \{0\} \cup [L]$ and all $h \in [H]$, we assume that $\sum_{l=1}^{L} \boldsymbol{w}_h^{(l)} \boldsymbol{\theta}_h^{(l)} = \boldsymbol{\theta}_h^{(0)}$.*

Unlike existing knowledge transfer, we did not assume that the source and target data are similar. This assumption implies that the parameters of the source tasks can be combined to approximate the parameters of the target task. It is not a restrictive assumption as it allows for flexibility in the relationship between the source and target tasks.

We detail the target (source) MDP model. Specifically, we consider the setting with $L$ source data generated from the episodic linear MDP (Puterman, 2014; Sutton, 2018) with the state space $\mathcal{S}$, action space $\mathcal{A}$ and horizon $H$. We assume $\mathcal{P}^{(l)} = \left\{ \mathcal{P}_h^{(l)} \right\}_{h=1}^{H}$ and the reward function $r^{(l)} = \left\{ r_h^{(l)} \right\}_{h=1}^{H}$ are specified as follows:

$$\mathcal{P}_h^{(l)}\left(x_{h+1} \mid x_h, a_h\right) = \left\langle \boldsymbol{\phi}\left(x_h, a_h\right), \boldsymbol{\mu}_h^{(l)}\left(x_{h+1}\right) \right\rangle, \tag{6}$$

$$\mathbb{E}\left[r_h^{(l)}\left(x_h, a_h\right) \mid x_h, a_h\right] = \left\langle \boldsymbol{\phi}\left(x_h, a_h\right), \boldsymbol{\theta}_h^{(l)} \right\rangle. \tag{7}$$

For $l \in [L], h \in [H]$, where $x_h$ and $a_h$ are the state and action in the time $h$, respectively, $\boldsymbol{\mu}_h^{(l)}$ 's are unknown measures over $\mathcal{S}$, and $\boldsymbol{\phi}$ is known feature map. The feature map can be thought of as the representation of relevant time-varying covariates. These equations define the transition and reward functions in the linear MDP, where the transition probabilities and expected rewards are linear in the feature map $\boldsymbol{\phi}$. This linearity assumption simplifies the model and allows for more efficient learning and knowledge transfer between tasks.

Given the $l$ th MDP for $l \in [L]$, a dataset $\mathcal{D} = \left\{ \left(x_h^{\tau(l)}, a_h^{\tau(l)}, r_h^{\tau(l)}\right) \right\}_{\tau,h,l=1}^{n^{(l)},H,L}$ is collected a priori where at each step $h \in [H]$ of each trajectory $\tau \in \left[n^{(l)}\right]$, the agent takes the action $a_h^{\tau(l)} \sim \pi_h^{(l)}\left(\cdot \mid x_h^{\tau(l)}\right)$ at the state $x_h^{\tau(l)}$, receives the reward $r_h^{\tau(l)} = r_h^{(l)}\left(x_h^{\tau(l)}, a_h^{\tau(l)}\right)$ satisfying Equation (7) and observes the next state $x_{h+1}^{\tau(l)} \sim \mathbb{P}_h^{(l)}\left(\cdot \mid x_h = x_h^{\tau(l)}, a_h = a_h^{\tau(l)}\right)$ satisfying Equation (6). The transition probabilities only depend on features specified in $\boldsymbol{\phi}(x, a)$. All trajectories in $\mathcal{D}^{(l)}$ for $l \in [L]$ are assumed to be independent. We impose no constraint on the behavior policies $\pi_h^{(l)}$ 's and allow them to vary across the $L$ sites. This means that the data collection process is flexible and can capture a variety of behaviors and situations in the source tasks. The independence assumption ensures that the data from different trajectories is not correlated, which simplifies the analysis and allows for a more straightforward application of statistical techniques.

For any policy $\pi = \{\pi_h\}_{h=1}^{H}$, we define the state value function $V_h^{\pi(l)} : \mathcal{S} \to \mathbb{R}$ and the action-value function (Q-function) $Q_h^{\pi(l)} : \mathcal{S} \times \mathcal{A} \to \mathbb{R}$ for the $l$ th site at each step $h \in [H]$ as:

$$V_h^{\pi(l)}(x) = \mathbb{E}_{\pi}^{(l)}\left[\sum_{t=h}^{H} r_t^{(l)}\left(x_t, a_t\right) \mid x_h = x\right], \tag{8}$$

$$Q_h^{\pi(l)}(x, a) = \mathbb{E}_{\pi}^{(l)}\left[\sum_{t=h}^{H} r_t^{(l)}\left(x_t, a_t\right) \mid x_h = x, a_h = a\right]. \tag{9}$$

Here the expectation $\mathbb{E}_{\pi}^{(l)}$ is taken for the randomness of the trajectory induced by $\pi$, which is obtained by taking the action $a_h \sim \pi_h\left(\cdot \mid x_h\right)$ at the state $x_h$ and observing the next state $x_{h+1} \sim \mathbb{P}_h^{(l)}\left(\cdot \mid x_h, a_h\right)$ at each step $h \in [H]$. Meanwhile, we fix $x_h = x \in \mathcal{S}$ in Equation (8) and $(x_h, a_h) = (x, a) \in \mathcal{S} \times \mathcal{A}$ in Equation (9). Bellman equation implies

$$V_h^{\pi(l)}(x) = \left\langle Q_h^{\pi(l)}(x, \cdot), \pi_h(\cdot \mid x) \right\rangle_{\mathcal{A}}, \quad Q_h^{\pi(l)}(x, a) = \left(\mathbb{B}_h^{(l)} V_{h+1}^{\pi(l)}\right)(x, a),$$

where $\langle \cdot, \cdot \rangle_{\mathcal{A}}$ is the inner product over $\mathcal{A}$, $\mathbb{B}_h^{(l)}$ is the Bellman operator for any function $f : \mathcal{S} \to \mathbb{R}$, with $\mathbb{E}^{(l)}$ taken with respect to the randomness of the reward $r_h^{(l)}\left(x_h, a_h\right)$ and next state $x_{h+1}$ where $x_{h+1} \sim \mathbb{P}_h^{(l)}\left(x_{h+1} \mid x_h, a_h\right)$.

We define the empirical mean squared Bellman error (MSBE) at each step $h \in [H]$ as

$$M_h(\boldsymbol{w}) = \sum_{\tau=1}^{K} \left( r_h^\tau + \widehat{V}_{h+1}\left(x_{h+1}^\tau\right) - \boldsymbol{\psi}\left(x_h^\tau, a_h^\tau\right)^\top \boldsymbol{w} \right)^2$$

to measure the performance of parameter $\boldsymbol{w}$. Correspondingly, we set

$$\left(\widehat{\mathbb{B}}_h \widehat{V}_{h+1}\right)(x, a) = \boldsymbol{\psi}(x, a)^\top \widehat{\boldsymbol{w}}_h, \quad \text{where} \quad \widehat{\boldsymbol{w}}_h = \operatorname*{argmin}_{\boldsymbol{w} \in \mathbb{R}^d} M_h(\boldsymbol{w}) + \lambda \cdot \|\boldsymbol{w}\|_2^2 \quad (10)$$

at each step $h \in [H]$. Here $\lambda > 0$ is the regularization parameter. Note that $\widehat{\boldsymbol{w}}_h$ has the closed form

$$\widehat{\boldsymbol{w}}_h = \Lambda_h^{-1} \left( \sum_{\tau=1}^{K} \boldsymbol{\psi}\left(x_h^\tau, a_h^\tau\right) \cdot \left( r_h^\tau + \widehat{V}_{h+1}\left(x_{h+1}^\tau\right) \right) \right), \quad (11)$$

where

$$\Lambda_h = \sum_{\tau=1}^{K} \boldsymbol{\psi}\left(x_h^\tau, a_h^\tau\right) \boldsymbol{\psi}\left(x_h^\tau, a_h^\tau\right)^\top + \lambda \cdot I. \quad (12)$$

Meanwhile, we construct $\Gamma_h$ based on $\mathcal{D}$ as

$$\Gamma_h(x, a) = \gamma \cdot \left( \boldsymbol{\psi}(x, a)^\top \Lambda_h^{-1} \boldsymbol{\psi}(x, a) \right)^{1/2}. \quad (13)$$

at each step $h \in [H]$. Here $\gamma > 0$ is the scaling parameter. In addition, we construct the value function and action-value function based on $\mathcal{D}$ as

$$\widehat{Q}_h(x, a) = \min\left\{ \bar{Q}_h(x, a), H - h + 1 \right\}^+, \quad \text{where} \quad \bar{Q}_h(x, a) = \left(\widehat{\mathbb{B}}_h \widehat{V}_{h+1}\right)(x, a) - \Gamma_h(x, a).$$

$$\widehat{V}_h(x) = \left\langle \widehat{Q}_h(x, \cdot), \widehat{\pi}_h(\cdot \mid x) \right\rangle_{\mathcal{A}}, \quad \text{where} \quad \widehat{\pi}_h(\cdot \mid x) = \operatorname*{arg\,max}_{\pi_h} \left\langle \widehat{Q}_h(x, \cdot), \pi_h(\cdot \mid x) \right\rangle_{\mathcal{A}}.$$

By Equation 8 and 10, for any function $V$, there exists $\bar{\boldsymbol{w}}_h^{(l)} \in \mathbb{R}^{d_1}$ such that

$$\left(\mathbb{B}_h^{(l)} V\right)(x, a) = \left\langle \boldsymbol{\phi}(x, a), \bar{\boldsymbol{\beta}}_h^{(l)} \right\rangle = \left\langle \boldsymbol{\psi}(x, a), \bar{\boldsymbol{w}}_h^{(l)} \right\rangle$$

where $\bar{\boldsymbol{\beta}}_h^{(l)} = \boldsymbol{\theta}_h^{(l)} + \int_{x' \in \mathcal{S}} \boldsymbol{\mu}_h^{(l)}(x') V(x') \, \mathrm{d}x'$. Therefore, the coefficients $\bar{\boldsymbol{\beta}}_h^{(l)}$ can be estimated through linear regression if the values $\left(\mathbb{B}_h^{(l)} V\right)(x, a)$ are known, which inspires us to derive the KT-RL algorithm. Without loss of generality, we assume the horizon length of all tasks is the same, denoted as $H$. We also assume that the trajectories in different tasks are independent. These definitions and equations are standard in reinforcement learning (Jin et al., 2021) and describe the value functions and the Bellman operator. The assumption that the coefficients can be estimated through linear regression is based on the linearity of the MDP and allows us to develop efficient algorithms for learning and transfer.

**Definition 2** (Compliance). *For a dataset $\mathcal{D} = \{(x_h^\tau, a_h^\tau, r_h^\tau)\}_{\tau, h=1}^{K, H}$, let $\mathbb{P}_{\mathcal{D}}$ be the joint distribution of the data collecting process. We say $\mathcal{D}$ is compliant with an underlying $\mathrm{MDP}(\mathcal{S}, \mathcal{A}, H, \mathcal{P}, r)$ if*

$$\mathbb{P}_{\mathcal{D}}\left( r_h^\tau = r', x_{h+1}^\tau = x' \mid \left\{ \left(x_h^j, a_h^j\right) \right\}_{j=1}^{\tau}, \left\{ \left(r_h^j, x_{h+1}^j\right) \right\}_{j=1}^{\tau-1} \right)$$

$$= \mathbb{P}\left( r_h(s_h, a_h) = r', s_{h+1} = x' \mid s_h = x_h^\tau, a_h = a_h^\tau \right). \quad (14)$$

*for all $r' \in [0, 1]$ and $x' \in \mathcal{S}$ at each step $h \in [H]$ of each trajectory $\tau \in [K]$. Here $\mathbb{P}$ on the right-hand side of Equation (14) is taken with respect to the underlying MDP.*

This definition ensures that the dataset is collected in a manner that is consistent with the underlying MDP. It guarantees that the data reflects the true dynamics of the environment and can be used for reliable learning and inference.

**Assumption 2** (Data Collecting Process). *The dataset $\mathcal{D}$ that the learner has access to is compliant with the underlying $\mathrm{MDP}(\mathcal{S}, \mathcal{A}, H, \mathcal{P}, r)$.*

This assumption is crucial for our analysis as it allows us to make meaningful conclusions about the performance of our algorithms based on the available data.

## 5 ALGORITHM

In the previous sections, we have discussed the problem formulation and related concepts. We move on to the algorithm in this section. Inspired by Jin et al. (2021) , we notice the key step is to construct estimates $\widehat{V}_h^{(0)}$ of $V_h^{(0)}$ and $\widehat{\mathbb{B}}_h^{(0)}\widehat{V}_{h+1}^{(0)}$ of $\mathbb{B}_h^{(0)}V_h^{(0)}$ based on $\{\mathcal{D}^{(l)}\}_{l=0}^L$ and the parameter estimator of $\{\mathcal{D}^{(l)}\}_{l=1}^L$. Pessimism plays an important role in the control of suboptimality. Define $\mathcal{D} = \cup_{l=0}^L \mathcal{D}_l$. We achieve pessimism by the notion of confidence bound $\Gamma_h$ as follows.

**Definition 3.** *We say* $\{\Gamma_h : \mathcal{S} \times \mathcal{A} \to \mathbb{R}\}_{h=1}^H$ *is a $\xi$-confidence bound of* $V = \{V_h\}_{h=1}^H$ *with respect to* $\mathbb{P}_{\mathcal{D}}$ *if the event:*

$$\mathcal{E}(V) = \left\{ \left| \left(\widehat{\mathbb{B}}_h^{(0)}V_{h+1}\right)(x,a) - \left(\mathbb{B}_h^{(0)}V_{h+1}\right)(x,a)\right| \leq \Gamma_h(x,a) \text{ for all } (x,a) \in \mathcal{S} \times \mathcal{A}, h \in [H] \right\}$$
(15)

*satisfies* $\mathbb{P}_{\mathcal{D}}\left(\mathcal{E}(V)\right) \geq 1 - \xi$. *Here the value functions* $V = \{V_h\}_{h=1}^H$ *and* $\{\Gamma_h\}_{h=1}^H$ *can depend on* $\mathcal{D}$.

This definition is crucial for quantifying the uncertainty in our estimates. By ensuring that the event $\mathcal{E}(V)$ occurs with high probability, we can control the suboptimality of our algorithm, as will be explained later. By definition, $\Gamma_h$ quantifies the approximation error of $\widehat{\mathbb{B}}_h^{(0)}V_{h+1}$ for $\mathbb{B}_h^{(0)}V_{h+1}$, which is important in eliminating the spurious correlation as discussed in Jin et al. (2021).

### 5.1 FOR SOURCE TASK:

We define the empirical mean squared Bellman error (MSBE) at each step $h \in [H]$ as:

$$M_h^{(l)}(\boldsymbol{\beta}) = \sum_{\tau=1}^{n^{(l)}} \left( r_h^{\tau(l)} + \widehat{V}_{h+1}^{(l)}\left(x_{h+1}^{\tau(l)}\right) - \boldsymbol{\phi}\left(x_h^{\tau(l)}, a_h^{\tau(l)}\right)^\top \boldsymbol{\beta}^{(l)} \right)^2.$$

Correspondingly, at each step $h \in [H]$, we set:

$$\left(\widehat{\mathbb{B}}_h^{(l)}\widehat{V}_{h+1}^{(l)}\right)(x,a) = \boldsymbol{\phi}(x,a)^\top \widehat{\boldsymbol{\beta}}_h^{(l)}, \quad \text{where} \quad \widehat{\boldsymbol{\beta}}_h^{(l)} = \arg\min_{\beta \in \mathbb{R}^d} M_h^{(l)}(\boldsymbol{\beta}) + \lambda \cdot \|\boldsymbol{\beta}\|_2^2.$$

Here $\lambda > 0$ is the regularization parameter. Note that $\widehat{\boldsymbol{\beta}}_h^{(l)}$ has the closed form

$$\widehat{\boldsymbol{\beta}}_h^{(l)} = \Lambda_h^{(l)^{-1}} \left( \sum_{\tau=1}^{n^{(l)}} \boldsymbol{\phi}\left(x_h^{\tau(l)}, a_h^{\tau(l)}\right) \cdot \left( r_h^{\tau(l)} + \widehat{V}_{h+1}^{(l)}\left(x_{h+1}^{\tau(l)}\right) \right) \right),$$

$$\text{where } \Lambda_h^{(l)} = \sum_{\tau=1}^{n^{(l)}} \boldsymbol{\phi}\left(x_h^{\tau(l)}, a_h^{\tau(l)}\right) \boldsymbol{\phi}\left(x_h^{\tau(l)}, a_h^{\tau(l)}\right)^\top + \lambda \cdot I.$$

Meanwhile, at each step $h \in [H]$, we construct $\Gamma_h^{(l)}$ based on $\mathcal{D}$ as:

$$\Gamma_h^{(l)}(x,a) = \gamma \cdot \left( \boldsymbol{\phi}(x,a)^\top \Lambda_h^{(l)^{-1}} \boldsymbol{\phi}(x,a) \right)^{1/2}$$

Here $\gamma > 0$ is the scaling parameter. In addition, we construct $\widehat{V}_h^{(l)}$ based on $\mathcal{D}$ as

$$\widehat{Q}_h^{(l)}(x,a) = \min\left\{ \bar{Q}_h^{(l)}(x,a), H - h + 1 \right\}^+, \text{where} \quad \bar{Q}_h^{(l)}(x,a) = \left(\widehat{\mathbb{B}}_h^{(l)}\widehat{V}_{h+1}^{(l)}\right)(x,a) - \Gamma_h^{(l)}(x,a).$$

$$\widehat{V}_h^{(l)}(x) = \left\langle \widehat{Q}_h^{(l)}(x,\cdot), \widehat{\pi}_h^{(l)}(\cdot \mid x) \right\rangle_{\mathcal{A}}, \quad \text{where} \quad \widehat{\pi}_h^{(l)}(\cdot \mid x) = \arg\max_{\pi_h} \left\langle \widehat{Q}_h^{(l)}(x,\cdot), \pi_h^{(l)}(\cdot \mid x) \right\rangle_{\mathcal{A}}.$$

In the source task, we use the MSBE to measure the error in estimating the Bellman operator. The closed form of $\widehat{\boldsymbol{\beta}}_h^{(l)}$ allows us to efficiently compute the estimate. The construction of $\Gamma_h^{(l)}$ and $\widehat{V}_h^{(l)}$ is based on the estimated Bellman operator and is designed to capture the uncertainty and optimize the policy.

## 5.2 FOR TARGET TASK:

The process for the target task is similar to before, we define the empirical mean squared Bellman error (MSBE) at each step $h \in [H]$ as:

$$M_h^{(0)}(\boldsymbol{w}) = \sum_{\tau=1}^{n^{(0)}} \left( r_h^{\tau(0)} + \widehat{V}_{h+1}^{(0)} \left( x_{h+1}^{\tau(0)} \right) - \boldsymbol{\psi} \left( x_h^{\tau(0)}, a_h^{\tau(0)} \right)^{\top} \boldsymbol{w} \right)^2.$$

Correspondingly, we set

$$\left( \widehat{\mathbb{B}}_h \widehat{V}_{h+1}^{(0)} \right)(x,a) = \psi(x,a)^{\top} \widehat{\boldsymbol{w}}_h, \quad \text{where} \quad \widehat{\boldsymbol{w}}_h = \underset{\boldsymbol{w} \in \mathbb{R}^L}{\arg\min} M_h^{(0)}(\boldsymbol{w}) + \lambda \cdot \|\boldsymbol{w}\|_2^2.$$

Here $\lambda > 0$ is the regularization parameter. Note that $\widehat{\boldsymbol{w}}_h$ has the closed form

$$\widehat{\boldsymbol{w}}_h = \Lambda_h^{(0)^{-1}} \left( \sum_{\tau=1}^{n^{(0)}} \boldsymbol{\psi} \left( x_h^{\tau(0)}, a_h^{\tau(0)} \right) \cdot \left( r_h^{\tau(0)} + \widehat{V}_{h+1}^{(0)} \left( x_{h+1}^{\tau(0)} \right) \right) \right), \qquad (16)$$

where $\Lambda_h^{(0)} = \sum_{\tau=1}^{n^{(l)}} \boldsymbol{\psi} \left( x_h^{\tau}, a_h^{\tau} \right) \boldsymbol{\psi} \left( x_h^{\tau}, a_h^{\tau} \right)^{\top} + \lambda \cdot I$. Meanwhile, we construct $\Gamma_h^{(0)}$ based on $\mathcal{D}$ as

$$\Gamma_h^{(0)}(x,a) = \gamma \cdot \left( \boldsymbol{\psi}(x,a)^{\top} \Lambda_h^{(0)^{-1}} \boldsymbol{\psi}(x,a) \right)^{1/2}.$$

Here $\gamma > 0$ is the scaling parameter. In addition, we construct $\widehat{V}_h^{(0)}$ based on $\mathcal{D}$ as

$$\widehat{Q}_h^{(0)}(x,a) = \min \left\{ \bar{Q}_h^{(0)}(x,a), H - h + 1 \right\}^+, \text{where } \bar{Q}_h^{(0)}(x,a) = \left( \widehat{\mathbb{B}}_h^{(0)} \widehat{V}_{h+1}^{(0)} \right)(x,a) - \Gamma_h^{(0)}(x,a).$$

$$\widehat{V}_h^{(0)}(x) = \left\langle \widehat{Q}_h^{(0)}(x,\cdot), \widehat{\pi}_h^{(0)}(\cdot \mid x) \right\rangle_{\mathcal{A}}, \text{ where } \quad \widehat{\pi}_h^{(0)}(\cdot \mid x) = \underset{\pi_h}{\arg\max} \left\langle \widehat{Q}_h^{(0)}(x,\cdot), \pi_h^{(0)}(\cdot \mid x) \right\rangle_{\mathcal{A}}.$$

For the target task, we define the MSBE similarly to the source task. The estimation of $\widehat{\boldsymbol{w}}_h$ and the construction of $\Gamma_h^{(0)}$ and $\widehat{V}_h^{(0)}$ are also based on the corresponding data and aim to optimize the performance in the target task.

The specific algorithm procedure is summarized in Algorithm 1.

# 6 THEORETICAL RESULTS

The following theorem characterizes the suboptimality of Algorithm 1, which is defined in Equation (4).

**Theorem 1** (Suboptimality). *Suppose Assumption 2 holds and the underlying MDP is linear. In Algorithm 2, we set*

$$\lambda = 1, \quad \gamma = c \cdot LH\sqrt{\zeta}, \quad where \ \zeta = \log(2LHK/\xi).$$

*Here $c > 0$ is an absolute constant and $\xi \in (0,1)$ is the confidence parameter. The following statements hold: (i) $\left\{ \Gamma_h^{(0)} \right\}_{h=1}^{H}$ in Algorithm 1, which is specified in Equation (13), is a $\xi$-uncertainty quantifier, and hence (ii) under $\mathcal{E}$ defined in Equation (15), which satisfies $\mathbb{P}_{\mathcal{D}}(\mathcal{E}) \geq 1 - \xi$, for any $x \in \mathcal{S}$, Pess $(\mathcal{D})$ in Algorithm 1 satisfies*

$$\mathrm{SubOpt}(\mathrm{Pess}(\mathcal{D}); x) \leq 2\gamma \sum_{h=1}^{H} \mathbb{E}_{\pi^*} \left[ \left( \psi(s_h, a_h)^{\top} \Lambda_h^{(0)-1} \psi(s_h, a_h) \right)^{1/2} \mid s_1 = x \right].$$

*Here $\mathbb{E}_{\pi^*}$ is concerning the trajectory induced by $\pi^*$ in the underlying MDP given the fixed matrix $\Lambda_h$.*

This theorem provides an upper bound on the suboptimality of the algorithm. The term $\gamma$ and the expectation inside the summation quantify the deviation from the optimal policy. The result shows that by carefully choosing the parameters and ensuring the uncertainty quantifier condition, we can

---

**Algorithm 1** Knowledge Transfer for Offline Reinforcement Learning (KT-RL)

---

1: **Input**: Target samples $\mathcal{D}^{(0)} = \left\{ \left( x_h^{\tau(0)}, a_h^{\tau(0)}, r_h^{\tau(0)} \right) \right\}_{\tau,h=1}^{K,H}$; $L$ Source samples $\mathcal{D}^{(l)} = \left\{ \left( x_h^{\tau(l)}, a_h^{\tau(l)}, r_h^{\tau(l)} \right) \right\}_{\tau,h,l=1}^{K,H,L}$.

   **Output**: $\left\{ \widehat{\pi}_h^{(0)} \right\}_{h=1}^{H}$.

2: **Transferring Step:**

3: Initialization: Set $\widehat{V}_{H+1}^{(l)}(\cdot) \leftarrow 0$.

4: **for** l=1,2,..., L **do**

5:   **for** step $h = H, H-1, \ldots, 1$ **do**

6:     Set $\Lambda_h^{(l)} \leftarrow \sum_{\tau=1}^{n^{(l)}} \phi\left(x_h^\tau, a_h^\tau\right) \phi\left(x_h^\tau, a_h^\tau\right)^\top + \lambda \cdot I$.

7:     Set $\widehat{\boldsymbol{\theta}}_h^{(l)} \leftarrow \Lambda_h^{(l)-1} \left( \sum_{\tau=1}^{n^{(l)}} \phi\left(x_h^\tau, a_h^\tau\right) \cdot \left( r_h^{\tau(l)} + \widehat{V}_{h+1}^{(l)} \left( x_{h+1}^{\tau(l)} \right) \right) \right)$.

8:     Set $\bar{\Gamma}_h^{(l)}(\cdot, \cdot) \leftarrow \eta \cdot \left( \phi(\cdot, \cdot)^\top \Lambda_h^{(l)-1} \phi(\cdot, \cdot) \right)^{1/2}$.

9:   **end for**

10: **end for**

11: Set $\boldsymbol{\psi}(x_t^{\tau(0)}, a_t^{\tau(0)}) = \sum_{l=1}^L \widehat{\boldsymbol{w}}_h^{(l)} \boldsymbol{\phi}(x_t^{\tau(0)}, a_t^{\tau(0)})$;

12: Set $\Lambda_h^{(0)} \leftarrow \sum_{\tau=1}^{n^{(0)}} \boldsymbol{\psi}\left(x_h^{\tau(0)}, a_h^{\tau(0)}\right) \boldsymbol{\psi}\left(x_h^{\tau(0)}, a_h^{\tau(0)}\right)^\top + \lambda \cdot I$;

13: **Calibration Step:**

$$[\hat{w}^{(1)}, \cdots, \hat{w}^{(L)}] = \Lambda_h^{(0)-1} \left( \sum_{\tau=1}^{n^{(0)}} \boldsymbol{\psi}\left(x_h^\tau, a_h^\tau\right) \cdot \left( r_h^{\tau(0)} + \widehat{V}_{h+1}^{(0)} \left( x_{h+1}^{\tau(0)} \right) \right) \right);$$

14: Set $\hat{\boldsymbol{\beta}}^0 = \sum_{l=1}^L \widehat{\boldsymbol{w}}^{(l)} \hat{\boldsymbol{\beta}}^{(l)}$;

15: $\bar{Q}_h^{(0)}(\cdot, \cdot) \leftarrow \boldsymbol{\phi}(\cdot, \cdot)^\top \hat{\boldsymbol{\beta}}_h^{(0)} - \Gamma_h^{(0)}(\cdot, \cdot)$; {Pessimism}

16: $\widehat{Q}_h^{(0)}(\cdot, \cdot) \leftarrow \min\left\{ \bar{Q}_h^{(0)}(\cdot, \cdot), H-h+1 \right\}^+$; {Truncation}

17: $\widehat{\pi}_h^{(0)}(\cdot \mid \cdot) \leftarrow \arg\max_{\pi_h} \left\langle \widehat{Q}_h^{(0)}(\cdot, \cdot), \pi_h^{(0)}(\cdot \mid \cdot) \right\rangle_{\mathcal{A}}$;   {Optimization}

18: $\widehat{V}_h^{(0)}(\cdot) \leftarrow \left\langle \widehat{Q}_h^{(0)}(\cdot, \cdot), \widehat{\pi}_h^{(0)}(\cdot \mid \cdot) \right\rangle_{\mathcal{A}}$.   {Evaluation}

---

control the suboptimality of the algorithm. The result depends on the number of source tasks $L$ instead of the dimension $d$ (Jin et al., 2021). When we adjust the relevant quantity from what might be similar to $d$ into our $L$, if usually $L < d$, then in this upper bound expression, because the value of $L$ is relatively smaller, in the summation and related calculations, the value of the upper bound will be relatively smaller. This means that our estimation of the algorithm's suboptimality is more precise and the upper bound is tighter.

We highlight the following aspects of Theorem 1:

**Corollary 1** (Suboptimality of KT-RL with Well-Explored Dataset). *Suppose $\mathcal{D}$ consists of $K$ trajectories $\{(x_h^\tau, a_h^\tau, r_h^\tau)\}_{\tau,h=1}^{K,H}$ independently and identically induced by a fixed behavior policy $\bar{\pi}$ in the linear MDP. Meanwhile, suppose there exists an absolute constant $\underline{c} > 0$ such that*

$$\lambda_{\min}\left(\Sigma_h^{(0)}\right) \geq \underline{c}/L, \quad \text{where } \Sigma_h^{(0)} = \mathbb{E}_{\bar{\pi}}\left[\boldsymbol{\psi}\left(s_h, a_h\right) \boldsymbol{\psi}\left(s_h, a_h\right)^\top\right]$$

*at each step $h \in [H]$. Here $\mathbb{E}_{\bar{\pi}}$ is taken with respect to the trajectory induced by $\bar{\pi}$ in the underlying MDP. In Algorithm 1, we set*

$$\lambda = 1, \quad \gamma = c \cdot LH\sqrt{\zeta}, \quad \text{where } \zeta = \log(4LHK/\xi).$$

*Here $c > 0$ is an absolute constant and $\xi \in (0, 1)$ is the confidence parameter. Suppose we have $K \geq C \cdot d\log(4dH/\xi)$, where $C > 0$ is a sufficiently large absolute constant that depends on $\underline{c}$. For $\text{Pess}(\mathcal{D})$ in Algorithm 1, the event*

$$\mathcal{E}^* = \left\{ \text{SubOpt}(\text{Pess}(\mathcal{D}); x) \leq c' \cdot L^{3/2} H^2 K^{-1/2} \sqrt{\zeta} \text{ for all } x \in \mathcal{S} \right\}$$

*satisfies $\mathbb{P}_{\mathcal{D}}\left(\mathcal{E}^*\right) \geq 1 - \xi$. Here $c' > 0$ is an absolute constant that only depends on $\underline{c}$ and $c$.*

This corollary provides a result for the case when the dataset is well-explored. It shows that under certain conditions on the dataset and the parameters, the suboptimality of the algorithm can be further reduced, approaching a desired bound. Similar to the Theorem 1, $L$ appears in the suboptimality upper bound expression. When $L < d$, the value of terms like $L^{3/2}$ will be smaller than when using $d$. This makes the suboptimality upper bound we obtain tighter, that is, the performance estimation of the algorithm in this case of a well-explored dataset is more accurate and the range of the upper bound is smaller.

### 6.1 MINIMAX OPTIMALITY: INFORMATION-THEORETIC LOWER BOUND

We establish the minimax optimality of Theorems 1 via the following information-theoretic lower bound.

**Theorem 2** (Information-Theoretic Lower Bound). *For the output* $\mathrm{Algo}(\mathcal{D})$ *of any algorithm, there exist a linear* $\mathrm{MDP}$ $\mathcal{M} = (\mathcal{S}, \mathcal{A}, H, \mathcal{P}, r)$, *an initial state* $x \in \mathcal{S}$, *and a dataset* $\mathcal{D}$, *which is compliant with* $\mathcal{M}$, *such that:*

$$
\mathbb{E}_{\mathcal{D}}\left[\frac{\mathrm{SubOpt}(\mathrm{Algo}(\mathcal{D}); x)}{\sum_{h=1}^{H} \mathbb{E}_{\pi^*}\left[\left(\boldsymbol{\psi}\left(s_h, a_h\right)^{\top} \Lambda_h^{(0)-1} \boldsymbol{\psi}\left(s_h, a_h\right)\right)^{1/2} \mid s_1 = x\right]}\right] \geq c,
$$

*where* $c > 0$ *is an absolute constant. Here* $\mathbb{E}_{\pi^*}$ *is taken according to the trajectory induced by* $\pi^*$ *in the underlying MDP given the fixed matrix* $\Lambda_h^{(0)}$. *Meanwhile,* $\mathbb{E}_{\mathcal{D}}$ *is taken for* $\mathbb{P}_{\mathcal{D}}$, *where* $\mathrm{Algo}(\mathcal{D})$ *and* $\Lambda_h^{(0)}$ *depend on* $\mathcal{D}$.

This theorem establishes a lower bound on the suboptimality of any algorithm. It shows that there is a fundamental limit to the performance of algorithms, and our proposed algorithm achieves a performance that is close to this limit, indicating its optimality in a minimax sense.

## 7 CONCLUSION

In conclusion, we have presented a novel knowledge transfer framework for offline reinforcement learning. This framework addresses the crucial challenges of data scarcity and high-dimensional feature spaces. By assuming a linear relationship between target and source data and comprehensively considering task relationships within the linear MDP framework, we have introduced innovative approaches to knowledge transfer. Our theoretical contributions include establishing relationships between the learning processes of target and source tasks and providing bounds for algorithm evaluation, demonstrating both the suboptimality upper bound and the minimax optimality. In algorithm design, we have focused on efficient source data processing and integration, along with ensuring dataset compliance and trajectory independence. Overall, our work significantly contributes to the advancement of offline reinforcement learning, offering a practical and theoretically sound solution for more efficient learning in various applications.

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

APPENDIX

You may include other additional sections here.

## A PROOF FOR LEMMA 1

**Lemma 1** ($\xi$-Uncertainty Quantifier for Linear MDP). *Suppose that Assumption 2 holds and the underlying MDP is a linear MDP. In Algorithm 1, we set*

$$\lambda = 1, \quad \gamma = c \cdot LH\sqrt{\zeta}, \quad \text{where } \zeta = \log(2LHK/\xi)$$

*Here $c > 0$ is an absolute constant and $\xi \in (0,1)$ is the confidence parameter. It holds that $\{\Gamma_h\}_{h=1}^H$ specified in Equation (13) are $\xi$-uncertainty quantifiers, where $\left\{\widehat{V}_{h+1}\right\}_{h=1}^H$ used in Equation (15) are obtained by Algorithm 1.*

**Proof for Lemma 1.** It suffices to show that under Assumption 2, the event $\mathcal{E}$ defined in Equation (15) satisfies $\mathbb{P}_{\mathcal{D}}(\mathcal{E}) \geq 1 - \xi$ with the $\xi$-uncertainty quantifiers $\{\Gamma_h\}_{h=1}^H$ defined in Equation (13). To this end, we upper bound the difference between $\left(\mathbb{B}_h\widehat{V}_{h+1}\right)(x,a)$ and $\left(\widehat{\mathbb{B}}_h\widehat{V}_{h+1}\right)(x,a)$ for all $h \in [H]$ and all $(x,a) \in \mathcal{S} \times \mathcal{A}$, where the Bellman operator $\mathbb{B}_h$ is defined in Equation (3), the estimated Bellman operator $\widehat{\mathbb{B}}_h$ is defined in Equation (10), and the estimated value function $\widehat{V}_{h+1}$ is constructed in Line 18 of Algorithm 1.

For any function $V : \mathcal{S} \to [0, H]$, Definition 1 ensures that $\mathbb{P}_h V$ and $\mathbb{B}_h V$ are linear in the feature map $\phi$ for all $h \in [H]$. To see this, note that Equation (5) implies

$$(\mathbb{P}_h V)(x,a) = \left\langle \phi(x,a), \int_{\mathcal{S}} V(x')\boldsymbol{\mu}_h(x')\,\mathrm{d}x' \right\rangle, \quad \forall (x,a) \in \mathcal{S} \times \mathcal{A}, \forall h \in [H]. \quad (17)$$

Also, Equation (5) ensures that the expected reward is linear in $\phi$ for all $h \in [H]$, which implies

$$(\mathbb{B}_h V)(x,a) = \langle \phi(x,a), \boldsymbol{\theta}_h \rangle + \left\langle \phi(x,a), \int_{\mathcal{S}} V(x')\boldsymbol{\mu}_h(x')\,\mathrm{d}x' \right\rangle, \quad \forall (x,a) \in \mathcal{S} \times \mathcal{A}, \forall h \in [H]. \quad (18)$$

Hence, there exists an unknown vector $\boldsymbol{\beta}_h \in \mathbb{R}^d$ such that

$$\left(\mathbb{B}_h\widehat{V}_{h+1}\right)(x,a) = \phi(x,a)^\top\boldsymbol{\beta}_h = \boldsymbol{\psi}(x,a)^\top\boldsymbol{w}_h, \quad \forall (x,a) \in \mathcal{S} \times \mathcal{A}, \quad \forall h \in [H]. \quad (19)$$

Recall the definition of $\widehat{\boldsymbol{w}}_h$ in Equation (11) and the construction of $\widehat{\mathbb{B}}_h\widehat{V}_{h+1}$ in Equation (10). The following lemma upper bounds the norms of $\boldsymbol{w}_h$ and $\widehat{\boldsymbol{w}}_h$, respectively.

**Lemma 2** (Bounded Weights of Value Functions). *Let $V_{\max} > 0$ be an absolute constant. For any function $V : \mathcal{S} \to [0, V_{\max}]$ and any $h \in [H]$, we have*

$$\|\boldsymbol{\beta}_h\| \leq (1 + V_{\max})\sqrt{d}, \quad \left\|\widehat{\boldsymbol{\beta}}_h\right\| \leq H^2KL\sqrt{d}/\lambda, \quad \|\widehat{\boldsymbol{w}}_h\| \leq H\sqrt{KL/\lambda}.$$

**Proof of Lemma 2.** For all $h \in [H]$, Equations (18) and (19) imply

$$\boldsymbol{\beta}_h = \boldsymbol{\theta}_h + \int_{\mathcal{S}} V(x')\boldsymbol{\mu}_h(x')\,\mathrm{d}x'.$$

By the triangle inequality and the fact that $\|\boldsymbol{\mu}_h(\mathcal{S})\| \leq \sqrt{d}$ in Definition 1 with the notation $\|\boldsymbol{\mu}_h(\mathcal{S})\| = \int_{\mathcal{S}} \|\boldsymbol{\mu}_h(x')\|\,\mathrm{d}x'$, we have

$$\|\boldsymbol{\beta}_h\| \leq \|\boldsymbol{\theta}_h\| + \left\|\int_{\mathcal{S}} V(x')\boldsymbol{\mu}_h(x')\,\mathrm{d}x'\right\| \quad (20)$$

$$\leq \|\boldsymbol{\theta}_h\| + \int_{\mathcal{S}} \|V(x')\boldsymbol{\mu}_h(x')\|\,\mathrm{d}x'$$

$$\leq \sqrt{d} + V_{\max} \cdot \|\mu_h(\mathcal{S})\|$$

$$\leq (1 + V_{\max})\sqrt{d}$$

where the third inequality follows from the fact that $V \in [0, V_{\max}]$. Meanwhile, by the definition of $\widehat{\boldsymbol{w}}_h$ in Equation (11) and the triangle inequality, we have

$$
\|\widehat{\boldsymbol{w}}_h\| = \left\| \Lambda_h^{-1} \left( \sum_{\tau=1}^{K} \boldsymbol{\psi}\left(x_h^\tau, a_h^\tau\right) \cdot \left(r_h^\tau + \widehat{V}_{h+1}\left(x_{h+1}^\tau\right)\right) \right) \right\|
$$

$$
\leq \sum_{\tau=1}^{K} \left\| \Lambda_h^{-1} \boldsymbol{\psi}\left(x_h^\tau, a_h^\tau\right) \cdot \left(r_h^\tau + \widehat{V}_{h+1}\left(x_{h+1}^\tau\right)\right) \right\|
$$

Note that $\left| r_h^\tau + \widehat{V}_{h+1}\left(x_{h+1}^\tau\right) \right| \leq H$, which follows from the fact that $r_h^\tau \in [0,1]$ and $\widehat{V}_{h+1} \in [0, H-1]$ by Line 18 of Algorithm 1. Also, note that $\Lambda_h \succeq \lambda \cdot I$, which follows from the definition of $\Lambda_h$ in Equation (12). Hence, we have

$$
\|\widehat{\boldsymbol{w}}_h\| \leq H \cdot \sum_{\tau=1}^{K} \left\| \Lambda_h^{-1} \boldsymbol{\psi}\left(x_h^\tau, a_h^\tau\right) \right\| = H \cdot \sum_{\tau=1}^{K} \sqrt{\boldsymbol{\psi}\left(x_h^\tau, a_h^\tau\right)^\top \Lambda_h^{-1/2} \Lambda_h^{-1} \Lambda_h^{-1/2} \boldsymbol{\psi}\left(x_h^\tau, a_h^\tau\right)}
$$

$$
\leq \frac{H}{\sqrt{\lambda}} \cdot \sum_{\tau=1}^{K} \sqrt{\boldsymbol{\psi}\left(x_h^\tau, a_h^\tau\right)^\top \Lambda_h^{-1} \boldsymbol{\psi}\left(x_h^\tau, a_h^\tau\right)}
$$

where the last inequality follows from the fact that $\left\| \Lambda_h^{-1} \right\|_{\mathrm{op}} \leq \lambda^{-1}$. Here $\|\cdot\|_{\mathrm{op}}$ denotes the matrix operator norm. By the Cauchy-Schwarz inequality, we have

$$
\|\widehat{\boldsymbol{w}}_h\| \leq H\sqrt{K/\lambda} \cdot \sqrt{\sum_{\tau=1}^{K} \psi\left(x_h^\tau, a_h^\tau\right)^\top \Lambda_h^{-1} \psi\left(x_h^\tau, a_h^\tau\right)}
$$

$$
= H\sqrt{K/\lambda} \cdot \sqrt{\mathrm{Tr}\left( \Lambda_h^{-1} \sum_{\tau=1}^{K} \psi\left(x_h^\tau, a_h^\tau\right) \psi\left(x_h^\tau, a_h^\tau\right)^\top \right)}
$$

$$
= H\sqrt{K/\lambda} \cdot \sqrt{\mathrm{Tr}\left( \Lambda_h^{-1}\left( \Lambda_h - \lambda \cdot I \right) \right)}
$$

$$
\leq H\sqrt{K/\lambda} \cdot \sqrt{\mathrm{Tr}\left( \Lambda_h^{-1} \Lambda_h \right)}
$$

$$
= H\sqrt{KL/\lambda} \tag{21}
$$

where the second equality follows from the definition of $\Lambda_h$ in Equation (12). Similarly, we can show that

$$
\left\| \widehat{\boldsymbol{\beta}}_h^{(l)} \right\| \leq H\sqrt{K/\lambda} \cdot \sqrt{\sum_{\tau=1}^{K} \boldsymbol{\phi}\left(x_h^\tau, a_h^\tau\right)^\top \Lambda_h^{(l)-1} \boldsymbol{\phi}\left(x_h^\tau, a_h^\tau\right)}
$$

$$
= H\sqrt{K/\lambda} \cdot \sqrt{\mathrm{Tr}\left( \Lambda_h^{(l)-1} \sum_{\tau=1}^{K} \boldsymbol{\phi}\left(x_h^\tau, a_h^\tau\right) \boldsymbol{\phi}\left(x_h^\tau, a_h^\tau\right)^\top \right)}
$$

$$
= H\sqrt{K/\lambda} \cdot \sqrt{\mathrm{Tr}\left( \Lambda_h^{(l)-1}\left( \Lambda_h^{(l)} - \lambda \cdot I \right) \right)}
$$

$$
\leq H\sqrt{K/\lambda} \cdot \sqrt{\mathrm{Tr}\left( \Lambda_h^{(l)-1} \Lambda_h^{(l)} \right)}
$$

$$
= H\sqrt{Kd/\lambda} \tag{22}
$$

$$
\left\| \widehat{\boldsymbol{\beta}}_h \right\| \leq \sum_{l=1}^{L} \left\| \widehat{\boldsymbol{\beta}}_h^{(l)} \boldsymbol{w}_h^{(l)} \right\| \leq \left\| [\widehat{\boldsymbol{\beta}}^{(1)}, \widehat{\boldsymbol{\beta}}^{(2)}, \ldots, \widehat{\boldsymbol{\beta}}^{(L)}] \right\| \|\boldsymbol{w}_h\| \leq \sqrt{L}H\sqrt{Kd/\lambda} \cdot H\sqrt{KL/\lambda} = H^2 KL\sqrt{d}/\lambda.
$$

Therefore, combining Equations (20) and (21), we conclude the proof of Lemma 2. We upper bound the difference between $\mathbb{B}_h \widehat{V}_{h+1}$ and $\widehat{\mathbb{B}}_h \widehat{V}_{h+1}$. For all $h \in [H]$ and all $(x, a) \in \mathcal{S} \times \mathcal{A}$, we have

$$\left(\mathbb{B}_h \widehat{V}_{h+1}\right)(x, a) - \left(\widehat{\mathbb{B}}_h \widehat{V}_{h+1}\right)(x, a) = \phi(x, a)^\top \left(\beta_h - \widehat{\beta}_h\right)$$

$$= \phi(x, a)^\top \beta_h - \psi(x, a)^\top \Lambda_h^{-1} \left(\sum_{\tau=1}^K \psi\left(x_h^\tau, a_h^\tau\right) \cdot \left(r_h^\tau + \widehat{V}_{h+1}\left(x_{h+1}^\tau\right)\right)\right)$$

$$= \underbrace{\phi(x, a)^\top \beta_h - \psi(x, a)^\top \Lambda_h^{-1} \left(\sum_{\tau=1}^K \psi\left(x_h^\tau, a_h^\tau\right) \cdot \left(\mathbb{B}_h \widehat{V}_{h+1}\right)\left(x_h^\tau, a_h^\tau\right)\right)}_{(i)}$$

$$\underbrace{- \psi(x, a)^\top \Lambda_h^{-1} \left(\sum_{\tau=1}^K \psi\left(x_h^\tau, a_h^\tau\right) \cdot \left(r_h^\tau + \widehat{V}_{h+1}\left(x_{h+1}^\tau\right) - \left(\mathbb{B}_h \widehat{V}_{h+1}\right)\left(x_h^\tau, a_h^\tau\right)\right)\right)}_{(ii)}. \quad (23)$$

Here the first equality follows from the definition of the Bellman operator $\mathbb{B}_h$ in Equation (3), the decomposition of $\mathbb{B}_h$ in Equation (19), and the definition of the estimated Bellman operator $\widehat{\mathbb{B}}_h$ in Equation (10), while the second equality follows from the definition of $\widehat{w}_h$. By the triangle inequality, we have

$$\left|\left(\mathbb{B}_h \widehat{V}_{h+1}\right)(x, a) - \left(\widehat{\mathbb{B}}_h \widehat{V}_{h+1}\right)(x, a)\right| \le |(i)| + |(ii)|.$$

In the sequel, we upper bound terms (i) and (ii) respectively. By the construction of the estimated value function $\widehat{V}_{h+1}$ in Line 18 of Algorithm 1, we have $\widehat{V}_{h+1} \in [0, H-1]$. By Lemma 2, we have $\|\beta_h\| \le H\sqrt{d}$. Hence, term (i) is upper bounded by

$$|(i)| = \left|\phi(x, a)^\top \beta_h - \psi(x, a)^\top \Lambda_h^{-1} \left(\sum_{\tau=1}^K \psi\left(x_h^\tau, a_h^\tau\right) \psi\left(x_h^\tau, a_h^\tau\right)^\top w_h\right)\right|$$

$$= \left|\psi(x, a)^\top w_h - \psi(x, a)^\top \Lambda_h^{-1} \left(\Lambda_h - \lambda \cdot I\right) w_h\right| = \lambda \cdot \left|\psi(x, a)^\top \Lambda_h^{-1} w_h\right|$$

$$\le \lambda \cdot \|w_h\|_{\Lambda_h^{-1}} \cdot \|\psi(x, a)\|_{\Lambda_h^{-1}} \le \lambda \cdot C\sqrt{1/\lambda} \cdot \sqrt{\psi(x, a)^\top \Lambda_h^{-1} \psi(x, a)}. \quad (24)$$

Here, we assume $\|w_h\|_{\Lambda_h^{-1}} < C$. Here the second equality follows from the definition of $\Lambda_h$ in Equation (12). Also, the first inequality follows from the Cauchy-Schwarz inequality, while the last inequality follows from the fact that

$$\|w_h\|_{\Lambda_h^{-1}} = \sqrt{w_h^\top \Lambda_h^{-1} w_h} \le \left\|\Lambda_h^{-1}\right\|_{op}^{1/2} \cdot \|w_h\| \le C\sqrt{1/\lambda}.$$

Here $\|\cdot\|_{op}$ denotes the matrix operator norm and we use the fact that $\left\|\Lambda_h^{-1}\right\|_{op} \le \lambda^{-1}$. It remains to upper bound term (ii). For notational simplicity, for any $h \in [H]$, any $\tau \in [K]$, and any function $V : \mathcal{S} \to [0, H]$, we define the random variable

$$\epsilon_h^\tau(V) = r_h^\tau + V\left(x_{h+1}^\tau\right) - \left(\mathbb{B}_h V\right)\left(x_h^\tau, a_h^\tau\right).$$

By the Cauchy-Schwarz inequality, term (ii) is upper bounded by

$$|(ii)| = \left|\psi(x, a)^\top \Lambda_h^{-1} \left(\sum_{\tau=1}^K \psi\left(x_h^\tau, a_h^\tau\right) \cdot \epsilon_h^\tau\left(\widehat{V}_{h+1}\right)\right)\right|$$

$$\le \left\|\sum_{\tau=1}^K \psi\left(x_h^\tau, a_h^\tau\right) \cdot \epsilon_h^\tau\left(\widehat{V}_{h+1}\right)\right\|_{\Lambda_h^{-1}} \cdot \|\psi(x, a)\|_{\Lambda_h^{-1}}$$

$$= \underbrace{\left\|\sum_{\tau=1}^K \psi\left(x_h^\tau, a_h^\tau\right) \cdot \epsilon_h^\tau\left(\widehat{V}_{h+1}\right)\right\|_{\Lambda_h^{-1}}}_{(iii)} \cdot \sqrt{\psi(x, a)^\top \Lambda_h^{-1} \psi(x, a)}. \quad (25)$$

In the sequel, we upper bound term (iii) via concentration inequalities. An obstacle is that $\widehat{V}_{h+1}$ depends on $\{(x_h^\tau, a_h^\tau)\}_{\tau=1}^K$ via $\{(x_{h'}^\tau, a_{h'}^\tau)\}_{\tau\in[K], h'>h}$, as it is constructed based on the dataset $\mathcal{D}$. To this end, we resort to uniform concentration inequalities to upper bound

$$\sup_{V\in\mathcal{V}_{h+1}(R,B,\lambda)} \left\| \sum_{\tau=1}^K \boldsymbol{\psi}\left(x_h^\tau, a_h^\tau\right)\cdot \epsilon_h^\tau(V) \right\|$$

for each $h\in[H]$, where it holds that $\widehat{V}_{h+1}\in\mathcal{V}_{h+1}(R,B,\lambda)$. Here for all $h\in[H]$, we define the function class

$$\mathcal{V}_h(R,B,\lambda) = \left\{ V_h(x;\boldsymbol{\theta},\gamma,\Sigma): \mathcal{S}\to[0,H] \text{ with } \|\theta\|\le R, \gamma\in[0,B], \Sigma\succeq\lambda\cdot I \right\}$$

where $V_h(x;\boldsymbol{\theta},\gamma,\Sigma) = \max_{a\in\mathcal{A}}\left\{ \min\left\{ \boldsymbol{\phi}(x,a)^\top\boldsymbol{\theta} - \gamma\cdot\sqrt{\boldsymbol{\psi}(x,a)^\top\Sigma^{-1}\boldsymbol{\psi}(x,a)}, H-h+1 \right\}^+ \right\}$.

For all $\varepsilon>0$ and all $h\in[H]$, let $\mathcal{N}_h(\varepsilon;R,B,\lambda)$ be the minimal $\varepsilon$-cover of $\mathcal{V}_h(R,B,\lambda)$ with respect to the supremum norm. In other words, for any function $V\in\mathcal{V}_h(R,B,\lambda)$, there exists a function $V^\dagger\in\mathcal{N}_h(\varepsilon;R,B,\lambda)$ such that

$$\sup_{x\in\mathcal{S}}\left|V(x) - V^\dagger(x)\right| \le \varepsilon.$$

Meanwhile, among all $\varepsilon$-covers of $\mathcal{V}_h(R,B,\lambda)$ defined by such a property, we choose $\mathcal{N}_h(\varepsilon;R,B,\lambda)$ as the one with the minimal cardinality. By Lemma 2, we have $\left\|\widehat{\boldsymbol{\beta}}_h\right\|\le H^2KL\sqrt{d}/\lambda$. Hence, for all $h\in[H]$, we have

$$\widehat{V}_{h+1}\in\mathcal{V}_{h+1}\left(R_0,B_0,\lambda\right), \quad \text{where } R_0 = H^2KL\sqrt{d}/\lambda, B_0 = 2\gamma.$$

Here $\lambda>0$ is the regularization parameter and $\gamma>0$ is the scaling parameter, which are specified in Algorithm 1. For notational simplicity, we use $\mathcal{V}_{h+1}$ and $\mathcal{N}_{h+1}(\varepsilon)$ to denote $\mathcal{V}_{h+1}\left(R_0,B_0,\lambda\right)$ and $\mathcal{N}_{h+1}\left(\varepsilon;R_0,B_0,\lambda\right)$, respectively. As it holds that $\widehat{V}_{h+1}\in\mathcal{V}_{h+1}$ and $\mathcal{N}_{h+1}(\varepsilon)$ is an $\varepsilon$-cover of $\mathcal{V}_{h+1}$, there exists a function $V_{h+1}^\dagger\in\mathcal{N}_{h+1}(\varepsilon)$ such that

$$\sup_{x\in\mathcal{S}}\left|\widehat{V}_{h+1}(x) - V_{h+1}^\dagger(x)\right| \le \varepsilon. \tag{26}$$

Hence, given $V_{h+1}^\dagger$ and $\widehat{V}_{h+1}$, the monotonicity of conditional expectations implies

$$\left|\left(\mathbb{P}_h V_{h+1}^\dagger\right)(x,a) - \left(\mathbb{P}_h\widehat{V}_{h+1}\right)(x,a)\right| \tag{27}$$

$$= \left|\mathbb{E}\left[V_{h+1}^\dagger\left(s_{h+1}\right) \mid s_h=x, a_h=a\right] - \mathbb{E}\left[\widehat{V}_{h+1}\left(s_{h+1}\right) \mid s_h=x, a_h=a\right]\right|$$

$$\le \mathbb{E}\left[\left|V_{h+1}^\dagger\left(s_{h+1}\right) - \widehat{V}_{h+1}\left(s_{h+1}\right)\right| \mid s_h=x, a_h=a\right] \le \varepsilon, \quad \forall(x,a)\in\mathcal{S}\times\mathcal{A}, \forall h\in[H].$$

Here the conditional expectation is induced by the transition kernel $\mathcal{P}_h(\cdot\mid x,a)$. Combining Equation (27) and the definition of the Bellman operator $\mathbb{B}_h$ in Equation (3), we have

$$\left|\left(\mathbb{B}_h V_{h+1}^\dagger\right)(x,a) - \left(\mathbb{B}_h\widehat{V}_{h+1}\right)(x,a)\right| \le \varepsilon, \quad \forall(x,a)\in\mathcal{S}\times\mathcal{A}, \forall h\in[H] \tag{28}$$

By the triangle inequality, Equations (26) and (28) imply

$$\left|\left(r_h(x,a) + \widehat{V}_{h+1}\left(x'\right) - \left(\mathbb{B}_h\widehat{V}_{h+1}\right)(x,a)\right) - \left(r_h(x,a) + V_{h+1}^\dagger\left(x'\right) - \left(\mathbb{B}_h V_{h+1}^\dagger\right)(x,a)\right)\right| \le 2\varepsilon \tag{29}$$

for all $h\in[H]$ and all $(x,a,x')\in\mathcal{S}\times\mathcal{A}\times\mathcal{S}$. Setting $(x,a,x') = \left(x_h^\tau, a_h^\tau, x_{h+1}^\tau\right)$ in Equation (29), we have

$$\left|\epsilon_h^\tau\left(\widehat{V}_{h+1}\right) - \epsilon_h^\tau\left(V_{h+1}^\dagger\right)\right| \le 2\varepsilon, \quad \forall\tau\in[K], \forall h\in[H] \tag{30}$$

Also, recall the definition of term (iii). By the Cauchy-Schwarz inequality, for any two vectors $a,b\in\mathbb{R}^d$ and any positive definite matrix $\Lambda\in\mathbb{R}_+^{d\times d}$, it holds that $\|a+b\|_\Lambda^2 \le 2\cdot\|a\|_\Lambda^2 + 2\cdot\|b\|_\Lambda^2$. Hence, for all $h\in[H]$, we have

$$|\text{(iii)}|^2 \le 2\cdot\left\|\sum_{\tau=1}^K \boldsymbol{\psi}\left(x_h^\tau, a_h^\tau\right)\cdot\epsilon_h^\tau\left(V_{h+1}^\dagger\right)\right\|_{\Lambda_h^{-1}}^2 + 2\cdot\left\|\sum_{\tau=1}^K \boldsymbol{\psi}\left(x_h^\tau, a_h^\tau\right)\cdot\left(\epsilon_h^\tau\left(\widehat{V}_{h+1}\right) - \epsilon_h^\tau\left(V_{h+1}^\dagger\right)\right)\right\|_{\Lambda_h^{-1}}^2 \tag{31}$$

The second term on the right-hand side is upper bounded by

$$2 \cdot \| \sum_{\tau=1}^{K} \boldsymbol{\psi} \left( x_h^\tau, a_h^\tau \right) \cdot \left( \epsilon_h^\tau \left( \widehat{V}_{h+1} \right) - \epsilon_h^\tau \left( V_{h+1}^\dagger \right) \right) \|_{\Lambda_h^{-1}}^2$$

$$= 2 \cdot \sum_{\tau, \tau'=1}^{K} \boldsymbol{\psi} \left( x_h^\tau, a_h^\tau \right)^\top \Lambda_h^{-1} \boldsymbol{\psi} \left( x_h^{\tau'}, a_h^{\tau'} \right) \cdot \left( \epsilon_h^\tau \left( \widehat{V}_{h+1} \right) - \epsilon_h^\tau \left( V_{h+1}^\dagger \right) \right) \cdot \left( \epsilon_h^{\tau'} \left( \widehat{V}_{h+1} \right) - \epsilon_h^{\tau'} \left( V_{h+1}^\dagger \right) \right)$$

$$\leq 8\varepsilon^2 \cdot \sum_{\tau, \tau'=1}^{K} \left| \boldsymbol{\psi} \left( x_h^\tau, a_h^\tau \right)^\top \Lambda_h^{-1} \boldsymbol{\psi} \left( x_h^{\tau'}, a_h^{\tau'} \right) \right| \leq 8\varepsilon^2 \cdot \sum_{\tau, \tau'=1}^{K} \| \boldsymbol{\psi} \left( x_h^\tau, a_h^\tau \right) \| \cdot \left\| \boldsymbol{\psi} \left( x_h^{\tau'}, a_h^{\tau'} \right) \right\| \cdot \| \Lambda_h^{-1} \|_{\mathrm{op}}$$

where the first inequality follows from Equation (30). As it holds that $\Lambda_h \succeq \lambda \cdot I$ by the definition of $\Lambda_h$ in Equation (12) and $\|\boldsymbol{\psi}(x,a)\| \leq \|\hat{\beta}_h\| \|\phi(x,a)\| \leq H^2 K L \sqrt{d}/\lambda$ for all $(x,a) \in \mathcal{S} \times \mathcal{A}$ by Definition 1 , for all $h \in [H]$, we have

$$2 \cdot \left\| \sum_{\tau=1}^{K} \boldsymbol{\psi} \left( x_h^\tau, a_h^\tau \right) \cdot \left( \epsilon_h^\tau \left( \widehat{V}_{h+1} \right) - \epsilon_h^\tau \left( V_{h+1}^\dagger \right) \right) \right\|_{\Lambda_h^{-1}}^2 \leq 8\varepsilon^2 H^4 K^3 L^2 d/\lambda^3. \tag{32}$$

Combining Equations (31) and (32), for all $h \in [H]$, we have

$$| ( \mathrm{iii} )|^2 \leq 2 \cdot \sup_{V \in \mathcal{N}_{h+1}(\varepsilon)} \left\| \sum_{\tau=1}^{K} \phi \left( x_h^\tau, a_h^\tau \right) \cdot \epsilon_h^\tau(V) \right\|_{\Lambda_h^{-1}}^2 + 8\varepsilon^2 H^4 K^3 L^2 d/\lambda^3. \tag{33}$$

Note that the right-hand side of Equation (33) does not involve the estimated value functions $\widehat{Q}_h$ and $\widehat{V}_{h+1}$, which are constructed based on the dataset $\mathcal{D}$. Hence, it allows us to upper bound the first term via uniform concentration inequalities. We utilize the following lemma to characterize the first term for any fixed function $V \in \mathcal{N}_{h+1}(\varepsilon)$. Recall the definition of $\epsilon_h^\tau(V)$. Also recall that $\mathbb{P}_\mathcal{D}$ is the joint distribution of the data collecting process.

**Lemma 3** (Concentration of Self-Normalized Processes). *Let $V : \mathcal{S} \to [0, H-1]$ be any fixed function. Under Assumption 2.2, for any fixed $h \in [H]$ and any $\delta \in (0,1)$, we have*

$$\mathbb{P}_\mathcal{D} \left( \left\| \sum_{\tau=1}^{K} \phi \left( x_h^\tau, a_h^\tau \right) \cdot \epsilon_h^\tau(V) \right\|_{\Lambda_h^{-1}}^2 > H^2 \cdot (2 \cdot \log(1/\delta) + d \cdot \log(1 + K/\lambda)) \right) \leq \delta.$$

**Proof of Lemma 3.** For the fixed $h \in [H]$ and all $\tau \in \{0, \ldots, K\}$, we define the $\sigma$-algebra

$$\mathcal{F}_{h,\tau} = \sigma \left( \left\{ \left( x_h^j, a_h^j \right) \right\}_{j=1}^{(\tau+1) \wedge K} \cup \left\{ \left( r_h^j, x_{h+1}^j \right) \right\}_{j=1}^{\tau} \right)$$

where $\sigma(\cdot)$ denotes the $\sigma$-algebra generated by a set of random variables and $(\tau+1) \wedge K$ denotes $\min\{\tau+1, K\}$. For all $\tau \in [K]$, we have $\phi(x_h^\tau, a_h^\tau) \in \mathcal{F}_{h,\tau-1}$, as $(x_h^\tau, a_h^\tau)$ is $\mathcal{F}_{h,\tau-1}$-measurable. Also, for the fixed function $V : \mathcal{S} \to [0, H-1]$ and all $\tau \in [K]$, we have

$$\epsilon_h^\tau(V) = r_h^\tau + V \left( x_{h+1}^\tau \right) - (\mathbb{B}_h V) \left( x_h^\tau, a_h^\tau \right) \in \mathcal{F}_{h,\tau}$$

as $\left( r_h^\tau, x_{h+1}^\tau \right)$ is $\mathcal{F}_{h,\tau}$-measurable. Hence, $\{\epsilon_h^\tau(V)\}_{\tau=1}^{K}$ is a stochastic process adapted to the filtration $\{\mathcal{F}_{h,\tau}\}_{\tau=0}^{K}$. We have

$$\mathbb{E}_\mathcal{D} \left[ \epsilon_h^\tau(V) \mid \mathcal{F}_{h,\tau-1} \right] = \mathbb{E}_\mathcal{D} \left[ r_h^\tau + V \left( x_{h+1}^\tau \right) \mid \left\{ \left( x_h^j, a_h^j \right) \right\}_{j=1}^{\tau}, \left\{ \left( r_h^j, x_{h+1}^j \right) \right\}_{j=1}^{\tau-1} \right] - (\mathbb{B}_h V) \left( x_h^\tau, a_h^\tau \right)$$

$$= \mathbb{E} \left[ r_h \left( s_h, a_h \right) + V \left( s_{h+1} \right) \mid s_h = x_h^\tau, a_h = a_h^\tau \right] - (\mathbb{B}_h V) \left( x_h^\tau, a_h^\tau \right) = 0$$

where the second equality follows from Equation (14) and the last equality follows from the definition of the Bellman operator $\mathbb{B}_h$. Here $\mathbb{E}_\mathcal{D}$ is taken with respect to $\mathbb{P}_\mathcal{D}$, while $\mathbb{E}$ is taken with respect to the immediate reward and next state in the underlying MDP. Moreover, as it holds that $r_h^\tau \in [0, 1]$ and $V \in [0, H-1]$, we have $r_h^\tau + V \left( x_{h+1}^\tau \right) \in [0, H]$. Meanwhile, we have $(\mathbb{B}_h V) \left( x_h^\tau, a_h^\tau \right) \in [0, H]$,

which implies $|\epsilon_h^\tau(V)| \le H$. Hence, for the fixed $h \in [H]$ and all $\tau \in [K]$, the random variable $\epsilon_h^\tau(V)$ is mean-zero and $H$-sub-Gaussian conditioning on $\mathcal{F}_{h,\tau-1}$.

We invoke Lemma E. 2 with $M_0 = \lambda \cdot I$ and $M_k = \lambda \cdot I + \sum_{\tau=1}^k \boldsymbol{\psi}(x_h^\tau, a_h^\tau) \boldsymbol{\psi}(x_h^\tau, a_h^\tau)^\top$ for all $k \in [K]$. For the fixed function $V : \mathcal{S} \to [0, H-1]$ and fixed $h \in [H]$, we have

$$\mathbb{P}_{\mathcal{D}} \left( \left\| \sum_{\tau=1}^K \boldsymbol{\psi}(x_h^\tau, a_h^\tau) \cdot \epsilon_h^\tau(V) \right\|_{\Lambda_h^{-1}}^2 > 2H^2 \cdot \log \left( \frac{\det(\Lambda_h)^{1/2}}{\delta \cdot \det(\lambda \cdot I)^{1/2}} \right) \right) \le \delta$$

for all $\delta \in (0, 1)$. Here we use the fact that $M_K = \Lambda_h$. Note that $\|\boldsymbol{\phi}(x, a)\| \le 1$ and $\|\boldsymbol{\psi}(x, a)\| \le \|\hat{\beta}_h\| \|\boldsymbol{\phi}(x, a)\| \le H^2 K L \sqrt{d}/\lambda$ for all $(x, a) \in \mathcal{S} \times \mathcal{A}$ by Definition 1. We have

$$\|\Lambda_h\|_{\mathrm{op}} = \left\| \lambda \cdot I + \sum_{\tau=1}^K \boldsymbol{\psi}(x_h^\tau, a_h^\tau) \boldsymbol{\psi}(x_h^\tau, a_h^\tau)^\top \right\|_{\mathrm{op}}$$

$$\le \lambda + \sum_{\tau=1}^K \left\| \boldsymbol{\psi}(x_h^\tau, a_h^\tau) \boldsymbol{\psi}(x_h^\tau, a_h^\tau)^\top \right\|_{\mathrm{op}} \le \lambda + H^4 K^3 L^2 d/\lambda^2$$

where $\| \cdot \|_{\mathrm{op}}$ denotes the matrix operator norm. Hence, it holds that $\det(\Lambda_h) \le (\lambda + H^4 K^3 L^2 d/\lambda^2)^L$ and $\det(\lambda \cdot I) = \lambda^L$, which implies

$$\mathbb{P}_{\mathcal{D}} \left( \left\| \sum_{\tau=1}^K \boldsymbol{\psi}(x_h^\tau, a_h^\tau) \cdot \epsilon_h^\tau(V) \right\|_{\Lambda_h^{-1}}^2 > H^2 \cdot (2 \cdot \log(1/\delta) + L \cdot \log(1 + H^4 K^3 L^2 d/\lambda^3)) \right)$$

$$\le \mathbb{P}_{\mathcal{D}} \left( \left\| \sum_{\tau=1}^K \boldsymbol{\psi}(x_h^\tau, a_h^\tau) \cdot \epsilon_h^\tau(V) \right\|_{\Lambda_h^{-1}}^2 > 2H^2 \cdot \log \left( \frac{\det(\Lambda_h)^{1/2}}{\delta \cdot \det(\lambda \cdot I)^{1/2}} \right) \right) \le \delta.$$

Therefore, we conclude the proof of Lemma 3. Applying Lemma 3 and the union bound, for any fixed $h \in [H]$, we have

$$\mathbb{P}_{\mathcal{D}} \left( \sup_{V \in \mathcal{N}_{h+1}(\varepsilon)} \left\| \sum_{\tau=1}^K \phi(x_h^\tau, a_h^\tau) \cdot \epsilon_h^\tau(V) \right\|_{\Lambda_h^{-1}}^2 > H^2 \cdot (2 \cdot \log(1/\delta) + L \cdot \log(1 + H^4 K^3 L^2 d/\lambda^3)) \right)$$

$$\le \delta \cdot |\mathcal{N}_{h+1}(\varepsilon)|.$$

For all $\xi \in (0, 1)$ and all $\varepsilon > 0$, we set $\delta = \xi / (H \cdot |\mathcal{N}_{h+1}(\varepsilon)|)$. Hence, for any fixed $h \in [H]$, it holds that

$$\sup_{V \in \mathcal{N}_{h+1}(\varepsilon)} \left\| \sum_{\tau=1}^K \phi(x_h^\tau, a_h^\tau) \cdot \epsilon_h^\tau(V) \right\|_{\Lambda_h^{-1}}^2 \le H^2 \cdot (2 \cdot \log(H \cdot |\mathcal{N}_{h+1}(\varepsilon)| / \xi) c) \tag{34}$$

with probability at least $1 - \xi/H$, which is taken with respect to $\mathbb{P}_{\mathcal{D}}$. Define $M = 2H^2 \cdot 2 \cdot \log(H \cdot |\mathcal{N}_{h+1}(\varepsilon)| / \xi) + 2H^2 L \log(1 + H^4 K^3 L^2 d/\lambda^3) + 8\varepsilon^2 H^4 K^3 L^2 d/\lambda^3$. Combining Equations (33) and (34), we have

$$\mathbb{P}_{\mathcal{D}} \left( \bigcap_{h \in [H]} \left\{ \left\| \sum_{\tau=1}^K \boldsymbol{\psi}(x_h^\tau, a_h^\tau) \cdot \epsilon_h^\tau \left( \widehat{V}_{h+1} \right) \right\|_{\Lambda_h^{-1}}^2 \le M \right\} \right) \ge 1 - \xi \tag{35}$$

which follows from the union bound. It remains to choose a proper $\varepsilon > 0$ and upper bound the $\varepsilon$-covering number $|\mathcal{N}_{h+1}(\varepsilon)|$. In the sequel, we set $\varepsilon = dH/K$ and $\lambda = 1$. By Equation (35), for all $h \in [H]$, it holds that

$$\left\| \sum_{\tau=1}^K \boldsymbol{\psi}(x_h^\tau, a_h^\tau) \cdot \epsilon_h^\tau \left( \widehat{V}_{h+1} \right) \right\|_{\Lambda_h^{-1}}^2 \le 2H^2 \cdot 2 \cdot \log(H \cdot |\mathcal{N}_{h+1}(\varepsilon)| / \xi) + 2H^2 L \cdot \log(1 + H^4 K^3 L^2 d) + 8H^6 K L^2 d^3$$

$$\tag{36}$$

with probability at least $1 - \xi$, which is taken with respect to $\mathbb{P}_{\mathcal{D}}$. To upper bound $|\mathcal{N}_{h+1}(\varepsilon)|$, we utilize the following lemma. Recall the definition of the function class $\mathcal{V}_h(R, B, \lambda)$. Also, recall that $\mathcal{N}_h(\varepsilon; R, B, \lambda)$ is the minimal $\varepsilon$-cover of $\mathcal{V}_h(R, B, \lambda)$ with respect to the supremum norm.

**Lemma 4** ($\varepsilon$-Covering Number (Jin et al., 2020))**.** *For all $h \in [H]$ and all $\varepsilon > 0$, we have*

$$\log|\mathcal{N}_h(\varepsilon; R, B, \lambda)| \le L \cdot \log(1 + 4R/\varepsilon) + L^2 \cdot \log\left(1 + 8L^{1/2}B^2/\left(\varepsilon^2\lambda\right)\right).$$

Recall that

$$\widehat{V}_{h+1} \in \mathcal{V}_{h+1}\left(R_0, B_0, \lambda\right), \quad \text{where } R_0 = H^2KL\sqrt{d}/\lambda, B_0 = 2\gamma, \lambda = 1, \gamma = c \cdot LH\sqrt{\zeta}$$

Here $c > 0$ is an absolute constant, $\xi \in (0, 1)$ is the confidence parameter, and $\zeta = \log(2LHK/\xi)$ is specified in Algorithm 1. Recall that $\mathcal{N}_{h+1}(\varepsilon) = \mathcal{N}_{h+1}\left(\varepsilon; R_0, B_0, \lambda\right)$ is the minimal $\varepsilon$-cover of $\mathcal{V}_{h+1} = \mathcal{V}_{h+1}\left(R_0, B_0, \lambda\right)$ with respect to the supremum norm. Applying Lemma 4 with $\varepsilon = dH/K$, we have

$$\log|\mathcal{N}_{h+1}(\varepsilon)| \le L \cdot \log\left(1 + 4L^{-1/2}K^{3/2}\right) + L^2 \cdot \log\left(1 + 32c^2 \cdot L^{1/2}K^2\zeta\right) \qquad (37)$$

$$\le L \cdot \log\left(1 + 4L^{1/2}K^2\right) + L^2 \cdot \log\left(1 + 32c^2 \cdot L^{1/2}K^2\zeta\right)$$

As it holds that $\zeta > 1$, we set $c \ge 1$ to ensure that the second term on the right-hand side of Equation (37) is the dominating term, where $32c^2 \cdot L^{1/2}K^2\zeta \ge 1$. Hence, we have

$$\log|\mathcal{N}_{h+1}(\varepsilon)| \le 2L^2 \cdot \log\left(1 + 32c^2 \cdot L^{1/2}K^2\zeta\right) \le 2L^2 \cdot \log\left(64c^2 \cdot L^{1/2}K^2\zeta\right) \qquad (38)$$

By Equations (36) and (38), for all $h \in [H]$, it holds that

$$\left\|\sum_{\tau=1}^{K} \psi\left(x_h^\tau, a_h^\tau\right) \cdot \epsilon_h^\tau\left(\widehat{V}_{h+1}\right)\right\|_{\Lambda_h^{-1}}^2 \qquad (39)$$

$$\le 2H^2 \cdot \left(2 \cdot \log(H/\xi) + 4L^2 \cdot \log\left(64c^2 \cdot L^{1/2}K^2\zeta\right) + \log(1 + H^4K^3L^2d) + 4H^4KL^2d^3\right)$$

with probability at least $1 - \xi$, which is taken with respect to $\mathbb{P}_\mathcal{D}$. Note that $\log(1 + K) \le \log(2K) \le \zeta$ and $\log\zeta \le \zeta$. Hence, we have

$$2 \cdot \log(H/\xi) + 4L^2 \cdot \log\left(L^{1/2}K^2\zeta\right) + \log(1 + H^4K^3L^2d) + 4H^4KL^2d^3$$

$$\le 2L^2 \cdot \log\left(LHK^4/\xi\right) + H^4K^3L^2d + 4L^2\zeta + 4H^4KL^2d^3 \le 18H^4K^3L^2d^3\zeta.$$

As it holds that $\zeta > 1$ and $\log\zeta \le \zeta$, Equation (39) implies

$$\left\|\sum_{\tau=1}^{K} \psi\left(x_h^\tau, a_h^\tau\right) \cdot \epsilon_h^\tau\left(\widehat{V}_{h+1}\right)\right\|_{\Lambda_h^{-1}}^2 \le L^2H^2\zeta \cdot \left(36H^2K^3L^2d^3 + 8 \cdot \log\left(64c^2\right)\right) \qquad (40)$$

We set $c \ge 1$ to be sufficiently large, which ensures that $36H^2K^3L^2d^3 + 8 \cdot \log\left(64c^2\right) \le c^2/4$ on the right-hand side of Equation (40). By Equations (25) and (40), for all $h \in [H]$, it holds that

$$|(\text{ii})| \le c/2 \cdot LH\sqrt{\zeta} \cdot \sqrt{\psi(x, a)^\top\Lambda_h^{-1}\psi(x, a)} = \gamma/2 \cdot \sqrt{\psi(x, a)^\top\Lambda_h^{-1}\psi(x, a)} \qquad (41)$$

with probability at least $1 - \xi$, which is taken with respect to $\mathbb{P}_\mathcal{D}$. By Equations (13), (23), (24), and (41), for all $h \in [H]$ and all $(x, a) \in \mathcal{S} \times \mathcal{A}$, it holds that

$$\left|\left(\mathbb{B}_h\widehat{V}_{h+1}\right)(x, a) - \left(\widehat{\mathbb{B}}_h\widehat{V}_{h+1}\right)(x, a)\right| \le (H\sqrt{d} + \gamma/2) \cdot \sqrt{\psi(x, a)^\top\Lambda_h^{-1}\psi(x, a)} \le \Gamma_h(x, a)$$

with probability at least $1 - \xi$, which is taken with respect to $\mathbb{P}_\mathcal{D}$. In other words, $\{\Gamma_h\}_{h=1}^{H}$ are $\xi$-uncertainty quantifiers. Therefore, we conclude the proof of Lemma 1.

## B  Proof for Theorem 1

It suffices to show that $\{\Gamma_h\}_{h=1}^H$ are $\xi$-uncertainty quantifiers, which are defined in Definition 3. In the following lemma 1, we prove that such a statement holds when the regularization parameter $\lambda > 0$ and scaling parameter $\beta > 0$ in Algorithm 1 are properly chosen.

As Lemma 1 proves that $\{\Gamma_h\}_{h=1}^H$ are $\xi$-uncertainty quantifiers, $\mathcal{E}$ satisfies $\mathbb{P}_{\mathcal{D}}(\mathcal{E}) \geq 1 - \xi$. Recall that $\mathbb{P}_{\mathcal{D}}$ is the joint distribution of the data collecting process. By specializing Theorem 1 to the linear MDP, we have

$$\mathrm{SubOpt}(\mathrm{Pess}(\mathcal{D}); x) \leq 2 \sum_{h=1}^H \mathbb{E}_{\pi^*} [\Gamma_h(s_h, a_h) \mid s_1 = x]$$

$$= 2\gamma \sum_{h=1}^H \mathbb{E}_{\pi^*} \left[ \left( \psi(s_h, a_h)^\top \Lambda_h^{-1} \psi(s_h, a_h) \right)^{1/2} \mid s_1 = x \right]$$

for all $x \in \mathcal{S}$ under $\mathcal{E}$. Here the last equality follows from Equation (13). Therefore, we conclude the proof of Theorem 1.

## C  Proof for Collorary 1

Proof of Corollary 1. For all $h \in [H]$ and all $\tau \in [K]$, we define the random matrices

$$Z_h = \sum_{\tau=1}^K A_h^\tau, \quad A_h^\tau = \psi(x_h^\tau, a_h^\tau) \psi(x_h^\tau, a_h^\tau)^\top - \Sigma_h$$

$$\text{where } \Sigma_h = \mathbb{E}_{\bar{\pi}} \left[ \psi(s_h, a_h) \psi(s_h, a_h)^\top \right]$$

For all $h \in [H]$ and all $\tau \in [K]$, we have $\mathbb{E}_{\bar{\pi}}[A_h^\tau] = 0$. Here $\mathbb{E}_{\bar{\pi}}$ is taken with respect to the trajectory induced by the fixed behavior policy $\bar{\pi}$ in the underlying MDP. As the $K$ trajectories in the dataset $\mathcal{D}$ are i.i.d., for all $h \in [H]$, $\{(x_h^\tau, a_h^\tau, r_h^\tau)\}_{\tau=1}^K$ are also i.i.d.. Hence, for all $h \in [H]$, $\{A_h^\tau\}_{\tau=1}^K$ are i.i.d. and centered.

We assume $\|\phi(x, a)\| \leq 1$ and $\|\psi(x, a)\| \leq C$ for all $(x, a) \in \mathcal{S} \times \mathcal{A}$. By Jensen's inequality, we have

$$\|\Sigma_h\|_{\mathrm{op}} \leq \mathbb{E}_{\bar{\pi}} \left[ \left\| \psi(s_h, a_h) \psi(s_h, a_h)^\top \right\|_{\mathrm{op}} \right] \leq C^2.$$

For any vector $v \in \mathbb{R}^d$ with $\|v\| = 1$, the triangle inequality implies

$$\|A_h^\tau v\| \leq \left\| \psi(x_h^\tau, a_h^\tau) \psi(x_h^\tau, a_h^\tau)^\top v \right\| + \|\Sigma_h v\| \leq \|v\| + \|\Sigma_h\|_{\mathrm{op}} \cdot \|v\| \leq 2C^2$$

Hence, for all $h \in [H]$ and all $\tau \in [K]$, we have

$$\|A_h^\tau\|_{\mathrm{op}} \leq 2C^2, \quad \left\| A_h^\tau (A_h^\tau)^\top \right\|_{\mathrm{op}} \leq \|A_h^\tau\|_{\mathrm{op}} \cdot \left\| (A_h^\tau)^\top \right\|_{\mathrm{op}} \leq 4C^4$$

As $\{A_h^\tau\}_{\tau=1}^K$ are i.i.d. and centered, for all $h \in [H]$, we have

$$\left\| \mathbb{E}_{\bar{\pi}} \left[ Z_h Z_h^\top \right] \right\|_{\mathrm{op}} = \left\| \sum_{\tau=1}^K \mathbb{E}_{\bar{\pi}} \left[ A_h^\tau (A_h^\tau)^\top \right] \right\|_{\mathrm{op}}$$

$$= K \cdot \left\| \mathbb{E}_{\bar{\pi}} \left[ A_h^1 (A_h^1)^\top \right] \right\|_{\mathrm{op}} \leq K \cdot \mathbb{E}_{\bar{\pi}} \left[ \left\| A_h^1 (A_h^1)^\top \right\|_{\mathrm{op}} \right] \leq 4KC^4$$

where the first inequality follows from Jensen's inequality. Similarly, for all $h \in [H]$ and all $\tau \in [K]$, as it holds that

$$\left\| (A_h^\tau)^\top A_h^\tau \right\|_{\mathrm{op}} \leq \left\| (A_h^\tau)^\top \right\|_{\mathrm{op}} \cdot \|A_h^\tau\|_{\mathrm{op}} \leq 4C^4$$

we have

$$\left\| \mathbb{E}_{\bar{\pi}} \left[ Z_h^\top Z_h \right] \right\|_{\mathrm{op}} \leq 4KC^4$$

For any fixed $h \in [H]$ and any $t \geq 0$, we have

$$\mathbb{P}_{\mathcal{D}} \left( \|Z_h\|_{\mathrm{op}} > t \right) = \mathbb{P}_{\mathcal{D}} \left( \left\| \sum_{\tau=1}^{K} A_h^\tau \right\|_{\mathrm{op}} > t \right) \leq 2L \cdot \exp\left( -\frac{t^2/2}{4KC^4 + 2t/3} \right)$$

For all $\xi \in (0,1)$, we set $t = \sqrt{10KC^4 \cdot \log(4LH/\xi)}$. When $K$ is sufficiently large so that $K \geq \frac{5 \cdot \log(4LH/\xi)}{C^4}$, we have $2t/(3C^4) \leq K$. Hence, for the fixed $h \in [H]$, we have

$$\mathbb{P}_{\mathcal{D}} \left( \|Z_h\|_{\mathrm{op}} \leq t \right) \geq 1 - 2L \cdot \exp\left( -t^2/(8KC^4 + 4t/3) \right)$$
$$\geq 1 - 2L \cdot \exp\left( -t^2/(10KC^4) \right) = 1 - \xi/(2H)$$

By the union bound, for all $h \in [H]$, it holds that

$$\|Z_h/K\|_{\mathrm{op}} = \left\| \frac{1}{K} \sum_{\tau=1}^{K} \psi(x_h^\tau, a_h^\tau) \psi(x_h^\tau, a_h^\tau)^\top - \Sigma_h \right\|_{\mathrm{op}} \leq \sqrt{10/K \cdot \log(4LH/\xi)}$$

with probability at least $1 - \xi/2$, which is taken with respect to $\mathbb{P}_{\mathcal{D}}$. By the definition of $Z_h$, we have

$$Z_h = \sum_{\tau=1}^{K} \psi(x_h^\tau, a_h^\tau) \psi(x_h^\tau, a_h^\tau)^\top - K \cdot \Sigma_h = (\Lambda_h - \lambda \cdot I) - K \cdot \Sigma_h$$

Recall that there exists an absolute constant $\underline{c} > 0$ such that $\lambda_{\min}(\Sigma_h) \geq \underline{c}/L$, which implies $\left\| \Sigma_h^{-1} \right\|_{\mathrm{op}} \leq L/\underline{c}$. When $K$ is sufficiently large so that $K \geq 40L/\underline{c} \cdot \log(4LH/\xi)$, for all $h \in [H]$, it holds that

$$\lambda_{\min}(\Lambda_h/K) = \lambda_{\min}(\Sigma_h + \lambda/K \cdot I + Z_h/K)$$
$$\geq \lambda_{\min}(\Sigma_h) - \|Z_h/K\|_{\mathrm{op}} \geq \underline{c}/L - \sqrt{10/K \cdot \log(4LH/\xi)} \geq \underline{c}/(2L)$$

Hence, for all $h \in [H]$, it holds that

$$\left\| \Lambda_h^{-1} \right\|_{\mathrm{op}} \leq (K \cdot \lambda_{\min}(\Lambda_h/K))^{-1} \leq 2L/(K \cdot \underline{c})$$

with probability at least $1 - \xi/2$ with respect to $\mathbb{P}_{\mathcal{D}}$, which implies

$$\sqrt{\psi(x,a)^\top \Lambda_h^{-1} \psi(x,a)} \leq \|\psi(x,a)\| \cdot \left\| \Lambda_h^{-1} \right\|_{\mathrm{op}}^{1/2} \leq c'' \sqrt{L/K}, \quad \forall(x,a) \in \mathcal{S} \times \mathcal{A}, \forall h \in [H]$$

Here we define the absolute constant $c'' = \sqrt{2C^2/\underline{c}}$ and use the fact that $\|\psi(x,a)\| \leq C$ for all $(x,a) \in \mathcal{S} \times \mathcal{A}$.

We define the event

$$\mathcal{E}_1^* = \left\{ \sqrt{\phi(x,a)^\top \Lambda_h^{-1} \phi(x,a)} \leq c'' \sqrt{d/K} \text{ for all } (x,a) \in \mathcal{S} \times \mathcal{A} \text{ and all } h \in [H] \right\}$$

We have $\mathbb{P}_{\mathcal{D}}\left(\mathcal{E}_1^*\right) \geq 1 - \xi/2$ for $K \geq 40L/\underline{c} \cdot \log(4LH/\xi)$. Also, we define the event

$$\mathcal{E}_2^* = \left\{ \mathrm{SubOpt}(\widehat{\pi}; x) \leq 2\gamma \cdot \sum_{h=1}^{H} \mathbb{E}_{\pi^*}\left[ \sqrt{\psi\left(s_h, a_h\right)^\top \Lambda_h^{-1} \psi\left(s_h, a_h\right)} \mid s_1 = x \right] \text{ for all } x \in \mathcal{S} \right\}$$

Here we set $\gamma = c \cdot LH\sqrt{\log(4LHK/\xi)}$. We have $\mathbb{P}_{\mathcal{D}}\left(\mathcal{E}_2^*\right) \geq 1 - \xi/2$. Hence, when $K$ is sufficiently large so that $K \geq 40/\underline{c} \cdot \log(4LH/\xi)$, on the event $\mathcal{E}^* = \mathcal{E}_1^* \cap \mathcal{E}_2^*$, we have

$$\mathrm{SubOpt}\left(\widehat{\pi}; x\right) \leq 2\gamma \cdot H \cdot c'' \sqrt{L/K} = c' \cdot L^{3/2} H^2 \sqrt{\log(4LHK/\xi)/K}, \quad \forall x \in \mathcal{S}$$

By the union bound, we have $\mathbb{P}_{\mathcal{D}}\left(\mathcal{E}^*\right) \geq 1 - \xi$ with $c' = 2c \cdot c''$, where $c'' = \sqrt{2C^2/\underline{c}}$ Therefore, we conclude the proof of Corollary 1.