# OpenReview forum: "Breaking through Data Scarcity:  Knowledge Transfer in Offline Reinforcement Learning"
_ICLR.cc/2025/Conference — ICLR 2025 Conference Withdrawn Submission_

### Official Review · Reviewer_GCRq · 2024-10-19

**Soundness:** 3
**Presentation:** 4
**Contribution:** 3
**Rating:** 5
**Confidence:** 2

**Summary:**

This paper investigates the challenging problem of knowledge transfer in reinforcement learning and its application to solving the issue of scarce datasets in offline reinforcement learning. Specifically, the author integrates feature extraction through feature engineering into the Bellman update process, mathematically modeling the extraction of features from source data that are similar to those in target data. From the perspective of algorithm design, this is an elegant end-to-end approach. However, the paper does not provide any experimental validation, which makes it hard to directly prove that the algorithm works in practice.

I think the author's idea interesting. If the author can address my questions and concerns, I will increase the score.

**Strengths:**

# Written
The author has defined the reinforcement process very clearly. If the author's paper aims to address theoretical issues in knowledge transfer of reinforcement learning domain, then this paper is undoubtedly well-written. However, if the author hopes to extend the algorithm to practical applications, it is recommended to refer to the most effective reinforcement learning algorithms for further improvement. In particular, I mentioned some algorithm design-related issues in the question session.

# Soundness

**Importance.** The author has investigated a critically important issue in reinforcement learning. Knowledge transfer poses challenges in reinforcement learning, even within a single environment, making it far from straightforward. Additionally, the problem studied by the author is offline reinforcement learning, thus the author's research simultaneously addresses the challenge of scarcity of data in offline reinforcement learning.

**Reasonability.** I believe the author's method is reasonable. The author employs kernel techniques to map the features of the target domain and source domain into a high-dimensional space, and filters out similar features between domains based on the similarity between matrices, thereby achieving data augmentation. Additionally, this method does not require pre-training any estimators and has a relatively rigorous mathematical foundation.

**Weaknesses:**

I consider the author's idea quite interesting, but it has not been designed with reference to the current state-of-the-art reinforcement learning algorithms. I have already raised questions related to the algorithm design during the question session. Furthermore, although the author's algorithm is theoretically sound, no experiments have been provided for verification. If the author's primary goal is to address theoretical issues, then even a simple demo experiment could be used to validate the theoretical claims.

Additionally, please see questions.

**Questions:**

# Algorithm

**Q1:** In line 16 of the pseudocode, truncating Q is reasonable under the offline setting, but why does the truncated bond have a relationship with the sequence length?

**Q2:** In line 17 of the pseudocode, it is suggested to use $\log \pi$ instead of $\pi$. $Q\log\pi$, known as actor-critic, has been proven superior both experimentally and theoretically in reinforcement learning.

**Q3:** When modeling the Bellman equation, you chose to use $V(s')$ instead of $Q(s', a')$, where $a'\sim\pi(\cdot|s')$. The advantage of this is that it avoids accessing out-of-distribution (OOD) actions. However, there are also disadvantages, such as infinitely approaching the behavior policy, which may prevent the current algorithm from surpassing the behavior policy. Could you consider introducing the expected regression from Implicit Q Learning, which could potentially allow the algorithm to learn better decisions than the behavior policy?

[1] Ilya Kostrikov, Ashvin Nair, Sergey Levine. Offline Reinforcement Learning with Implicit Q-Learning

# Theoretical Proof

I  am still reviewing the mathematical proofs and may have additional questions or comments late.

---

> ### Comment · Reviewer_GCRq · 2024-11-28
> **Official Comments by Reviewer GCRq**
>
> I still believe that the author's idea has innovative elements. However, the author has not replied to my questions, and thus my doubts remain unresolved. Consequently, my concern about the author's positioning of this paper, whether it is purely theoretical or also takes into account experimental testing, will not be fully addressed. Therefore, if the author still fails to respond to my questions, I will consider lowering my score or confidence.

---

### Official Review · Reviewer_8GXE · 2024-10-24

**Soundness:** 2
**Presentation:** 2
**Contribution:** 2
**Rating:** 5
**Confidence:** 2

**Summary:**

The paper provides theoretical insights into knowledge transfer from source tasks to a target task in offline reinforcement learning (RL) to address data scarcity.
The analysis is applicable to algorithms operating within the linear MDP framework, where the target data is assumed to be a linear combination of the source data.

**Strengths:**

1. The paper presents a thoughtful combination of offline reinforcement learning and transfer learning, with useful mechanisms for handling uncertainty and leveraging source tasks.
2. The paper offers a rigorous theoretical analysis of the proposed approach, KT-RL.

**Weaknesses:**

1. Lack of experimental evaluation: The authors could prepare synthetic experiments to demonstrate the practical applicability and robustness of KT-RL.
2. The related work could look into some recent related knowledge transfer techniques in offline RL like [1] [2].
3. Calculation of inverse Gram matrices Λ(l)−1 can be computationally expensive if the features or samples is large
causing performance bottlenecks in high-dimensional settings.

Suggestion:

Since the paper uses many theoretical notations, it is hard to keep track of them, which hampers readability. Providing a notation table would enhance the paper's clarity and accessibility.


References:

[1] Gangopadhyay, Briti, et al. "Integrating Domain Knowledge for handling Limited Data in Offline RL." arXiv preprint arXiv:2406.07041 (2024).
[2] Wang, Zhao, et al. "Augmenting Offline RL with Unlabeled Data." arXiv preprint arXiv:2406.07117 (2024).

**Questions:**

Q1. What does data scarcity mean in quantifiable terms? Can the authors define data scarcity?

Q2. How relevant is the linear combination of source data resulting in target data assumption for practical scenarios. Are there some examples of tasks following such assumption?

Q3. In line 126, the authors state, "source data closely resembles target data, our approach diverges from this assumption."
However, they also assume that the target data is a linear combination of the source data. Could the authors elaborate on how this differs from related work? Additionally,
w_h^l  in Assumption 1 has not been defined before

Q4. What is ψ in MSBE? This term has not been defined before.

Q5. How sensitive is the algorithm to the choice of hyperparameters such as λ, γ, η?

Q6. How does Theorem 2 differ from Theorem 4.7 in (Jin et al., 2021)?

---

### Official Review · Reviewer_kEVf · 2024-10-29

**Soundness:** 2
**Presentation:** 2
**Contribution:** 2
**Rating:** 6
**Confidence:** 2

**Summary:**

The paper presents a theoretical framework for knowledge transfer in offline RL. This study considers a linear MDP and investigates an algorithm for transferring value functions and policies from a source domain to a target domain. The upper bound on suboptimality and the minimax optimality are established.

**Strengths:**

- I am not aware of prior theoretical studies on knowledge transfer in offline RL. It is interesting to see that an upper bound on suboptimality can be demonstrated for knowledge transfer in offline RL.

**Weaknesses:**

- The relationship to prior work on transfer learning in offline RL is unclear. Is this the first study to establish an upper bound on suboptimality in offline RL transfer learning? If not, please provide a comparison, such as how tight the upper bound is relative to previous work.

- No experiments are presented to evaluate the proposed algorithm. If possible, please provide experimental results on applicable tasks, such as simple tasks with tabular settings.

- There is no clear connection to recent deep RL algorithms. I believe that algorithms based on successor features could be a promising approach for implementing the proposed method in ways applicable to real-world problems. Please offer insights on how to adapt the proposed algorithm for real-world applications.

**Questions:**

-  Please provide a comparison, such as how tight the upper bound is relative to previous work.
- If possible, please provide experimental results on applicable tasks, such as simple tasks with tabular settings.
-  Please offer insights on how to adapt the proposed algorithm for real-world applications.

---

### Official Review · Reviewer_naX6 · 2024-11-01

**Soundness:** 2
**Presentation:** 2
**Contribution:** 2
**Rating:** 3
**Confidence:** 3

**Summary:**

The authors propose a method for knowledge transfer in offline RL. In their setting, there are L source tasks and a target task which is a linear combination of the source tasks in the linear MDP. There is an abundance of the data in the source tasks, but the target task has only limited data. The authors propose an algorithm for knowledge transfer in this case and provide a theoretical analysis on the optimality of the algorithm.

**Strengths:**

- This paper studies the problems within the domain of offline reinforcement learning which is an active area of research and has a potential to have a large impact in the field.
- The authors consider a novel problem setting and the assumption where the target task is a linear combination of several source tasks.
- The authors are rigorous in their definitions and approach to studying the problem.

**Weaknesses:**

- My main concern about the paper is that while it is set to address a practical problem of data scarcity in offline reinforcement learning in the target domain, it completely lacks any experiments, even in toy domains. Given that the paper proposes a new algorithm to address the problem and motivates the need for an algorithm by some practical limitations, I believe that the experimental section is necessary even though the paper's focus is theoretical analysis.
- I am not completely convinced by the assumption of the paper where the target task is a linear combination of the source tasks. What are the examples when this is relevant? How common are such examples in reality? How can we know if this assumption holds for a given task / environment?
- I think the clarity of the paper could be improved. For example, the real problem formulation comes only at the end of page 4, and it would be better to understand the main setting and assumptions early in the paper, as, for example, saying at the beginning that "target data is a linear combination of source data" is not enough and it should be explained what it means precisely.
- In several places in the paper, the authors rely on the closed-form solutions. With what size of the dataset is this applicable computationally?

**Questions:**

I would like the authors to comment on the applicability and realism of the assumption on the target domain being a linear combination of the source domains.

---

### Official Review · Reviewer_Q1VE · 2024-11-02

**Soundness:** 2
**Presentation:** 1
**Contribution:** 2
**Rating:** 3
**Confidence:** 3

**Summary:**

This paper focuses on the problem of knowledge transfer in offline reinforcement learning. The paper specifically focuses on a linear MDP setting, where transition function and reward are defined as linear functions of a set of features. In the knowledge transfer setting, there is a large amount of data from *source* tasks, and limited data for the desired *target* task. The key assumption is that the linear reward parameters of the target task are a linear combination of the reward parameters of the source tasks.

The proposed algorithm, KT-RL, involves constructing an estimate of the value function under the target task, using data from the source tasks. The value function, Q-function, and uncertainty are computed via Bellman iteration. The estimates for the target task reward function parameters are estimated from the source task statistics. The resulting Q-values are then penalized under uncertainty.

**Strengths:**

This paper introduces an offline knowledge transfer setting that is tractable for theoretical analysis. By making clear assumptions about the structure of the source and target data, they are able to derive appropriate bounds on the accuracy of the estimated value functions. To my knowledge this analysis has not been done on the specific offline-data, knowledge transfer regime.

**Weaknesses:**

The assumptions described in the paper are limiting. The algorithm assumes access to a prior-known feature map phi(x,a), in which the transition functions and rewards are linear. A large challenge in many practical RL problems is to learn this feature map -- presumably, offline data from source tasks can help in learning this map itself (which is not covered under these assumptions).

The contributions noted in the introduction lack justification. For example, assuming linearity is framed as breaking an assumption, when it instead limits the practicality of the results. The same holds for the assumption about dataset compliance and trajectory independence. The authors argue that their method "enhances privacy preservation", but does not elaborate on how this may be achieved.

While it is OK for the paper to focus entirely on theoretical contributions, since the paper is framed as providing an algorithm (KT-RL), at least some form of didactic example of this algorithm would strengthen the contribution.

The writing in the paper can use work on improving clarity. Many terms are introduced which are not properly defined. In addition, especially Section 5 about the algorithm is hard to read, and does not clearly explain the algorithmic procedure, nor the justification behind why it is designed as such. Sections 5.1 and 5.2 contain many repeated sentences and equations. The terms "Calibration" and "Truncation" are used in Algorithm 1, but are not defined anywhere in the paper. It is often unclear which quantities are assumed to be known ahead of time, and which are aimed to be discovered by the algorithm.

**Questions:**

What is the relation between data scarcity (as motivated in the introduction) and the proposed algorithm? Additional details on the motivation of what this analysis is aimed to provide would strengthen the paper.

What is the motivation behind the particular definition of suboptimality used in equation 4? A connection between this and a desired goal would be helpful (i.e. minimizing regret, maximizing expected performance of a policy under the target task, etc).

Why is the writing in 5.1 and 5.2 duplicated? For example MSBE is defined multiple times. If the intent is to showcase the difference between the target and source tasks, such as the presence of 'w', it would be clearer to show the comparison side by side.  Algorithm 1 and the algorithm section in general can use significant clarity improvements to make it clear what each part of the algorithm aims to accomplish, and to justify each step. For example, the calibration, pessimism, and truncation steps are undefined.

The paper can in general benefit from less overstated language. For example, the abstract and introduction talk about "practical and efficient utilization for applications", yet this claim is not demonstrated in the paper. The introduction also talks about powerful function approximation, yet the paper does not describe any relation to them.

---

### Note · Authors · 2024-11-28

**Comment:**

We would like to express our sincere gratitude for the time and effort you have dedicated to reviewing our submission for ICLR 2025. After careful consideration, we have decided to withdraw our current manuscript.
We acknowledge that there are deficiencies and lack of clarity in Assumption 1. We have already made new progress based on this and are in the process of refinement. Additionally, there are some issues with unclear descriptions and word choices in the article, which we will revise. We also plan to conduct further experiments to enhance this work.
Once again, we truly appreciate the valuable time and suggestions from the reviewers. We will strive to improve the quality of our work and may consider resubmitting it in the future.

**Withdrawal Confirmation:**

I have read and agree with the venue's withdrawal policy on behalf of myself and my co-authors.